# Structure and function of *Plasmodium* actin II in the parasite mosquito stages

**Andrea J. Lopez**[1☯], **Maria Andreadaki**[2☯], **Juha Vahokoski**[1], **Elena Deligianni**[2], **Lesley J. Calder**[3], **Serena Camerini**[4], **Anika Freitag**[2], **Ulrich Bergmann**[5], **Peter B. Rosenthal**[3], **Inga Sidén-Kiamos**[2]*, **Inari Kursula**[1,5]*

**1** Department of Biomedicine, University of Bergen, Bergen, Norway, **2** Institute of Molecular Biology and Biotechnology, Foundation for Research and Technology-Hellas, Heraklion, Greece, **3** Structural Biology of Cells and Viruses Laboratory, Francis Crick Institute, London, United Kingdom, **4** Istituto Superiore di Sanità, Roma, Italy, **5** Faculty of Biochemistry and Molecular Medicine, University of Oulu, Oulu, Finland

☯ These authors contributed equally to this work.
* inga@imbb.forth.gr (ISK); inari.kursula@uib.no (IK)

**Data Availability Statement:** The atomic coordinates have been deposited at the Protein Data Bank (PDB) with accession codes 8CCN and 8CCO. The density maps have been deposited at

## Abstract

Actins are filament-forming, highly-conserved proteins in eukaryotes. They are involved in essential processes in the cytoplasm and also have nuclear functions. Malaria parasites (*Plasmodium* spp.) have two actin isoforms that differ from each other and from canonical actins in structure and filament-forming properties. Actin I has an essential role in motility and is fairly well characterized. The structure and function of actin II are not as well understood, but mutational analyses have revealed two essential functions in male gametogenesis and in the oocyst. Here, we present expression analysis, high-resolution filament structures, and biochemical characterization of *Plasmodium* actin II. We confirm expression in male gametocytes and zygotes and show that actin II is associated with the nucleus in both stages in filament-like structures. Unlike actin I, actin II readily forms long filaments *in vitro*, and near-atomic structures in the presence or absence of jasplakinolide reveal very similar structures. Small but significant differences compared to other actins in the openness and twist, the active site, the D-loop, and the plug region contribute to filament stability. The function of actin II was investigated through mutational analysis, suggesting that long and stable filaments are necessary for male gametogenesis, while a second function in the oocyst stage also requires fine-tuned regulation by methylation of histidine 73. Actin II polymerizes *via* the classical nucleation-elongation mechanism and has a critical concentration of ~0.1 µM at the steady-state, like actin I and canonical actins. Similarly to actin I, dimers are a stable form of actin II at equilibrium.

## Author summary

Malaria is a parasitic infection caused by *Plasmodium* spp., which belong to the phylum Apicomplexa. In 2020, approximately 627000 people died from nearly 241 million malaria cases registered worldwide. In contrast to other apicomplexan parasites, *Plasmodium* spp. have two actin isoforms. While actin I is part of the glideosome complex, essential for

the Electron Microscopy Data Bank (EMDB) with accession codes EMD-10588 and EMD-10589. The mass spectrometry proteomics data have been deposited to the ProteomeXchange Consortium via the PRIDE partner repository with the dataset identifier PXD033378 and 10.6019/PXD033378.

**Funding:** This study was funded by grants from the Academy of Finland (310917 to IK), the Norwegian Research Council (262476 to IK), Sigrid Jusélius Foundation (to IK), Emil Aaltonen Foundation (to IK), Jane and Aatos Erkko Foundation (to IK), Fondation Sante, Greece (to ISK), State Scholarship Foundation IKY, Greece (MIS:5001552 to MA). PBR was supported by the Francis Crick Institute, which receives its core funding from Cancer Research UK (CC2106; PBR), the UK Medical Research Council (CC2106; PBR), and the Wellcome Trust (CC2106; PBR). The funders had no role in the study design, data collection and analysis, decision to publish, or preparation of the manuscript.

**Competing interests:** The authors have declared that no competing interests exist.

locomotion, actin II is present in the mosquito stages, where it has functions in gameto-genesis and in the oocyst. The exact function of actin II is still unclear, and information at the molecular level is limited. We show here that actin II is associated with the nucleus in gametocytes and zygotes and performs specific functions requiring both long filaments and fine-tuning by methylation of histidine 73. We determined the structures of the filamentous form of actin II at near-atomic resolution and characterized its polymerization properties *in vitro*. Our study provides a molecular basis for the differences between actins of the malaria parasite and humans, as well as between the two parasite actin isoforms. Furthermore, the *in vivo* studies provide insights into the function of actin II in the parasite.

## Introduction

The protozoan phylum Apicomplexa consists of more than 6000 parasitic species [1]. This group includes *Plasmodium* spp., well known as causative agents of human malaria, as well as several parasites responsible for diseases in other animals of socioeconomic importance [2]. The life cycle of malaria parasites involves several stages that go through different morphological states and infect different cell types in the mosquito and the vertebrate host.

Actin filaments are essential for parasite growth, development, motility, and host cell invasion [3,4]. Unlike other apicomplexan parasites, including *Toxoplasma gondii*, which have only a single actin isoform, there are two actin isoforms (actin I and actin II) in *Plasmodium*. Actin II is one of the most divergent actins among eukaryotes [5]. *Plasmodium falciparum* actin I shares 82–83% and actin II only 76–77% of common sequence with canonical actins from opisthokonts. The two actin isoforms have 79% sequence identity with each other [6]. While the function of actin I in development, motility, and invasion has been intensively studied [7,8], less is known about the functions of actin II, which is specific for gametogenesis and insect stages [9,10].

Mutants lacking actin II have a pleiotropic phenotype during the mosquito stages. Actin II is necessary for egress of the male gametocyte (microgametocyte) from the red blood cell and is strictly required for exflagellation of male gametes [10]. In the mutants, motile ookinetes are normally formed, but the protein has a role in the female gamete or zygote that manifests itself as a block of oocyst development in the absence of actin II. Complementation of an actin II deletion mutant with actin I resulted in poor restoration of exflagellation. Some ookinetes were recovered, and these traversed the midgut epithelium and formed oocysts, which however did not develop [9]. Furthermore, complementation of the deletion mutant with a gene expressing an actin I chimera, where the DNase I-binding (D-) loop in subdomain 2 was replaced by the muscle actin sequence, restored exflagellation [5].

The biological functions of actins are dependent on transitions between the globular G-form and the filamentous F-form, which are tightly coupled to ATP hydrolysis [11,12]. The structure of monomeric G-actin is well known; there are almost 200 crystal structures available in the Protein Data Bank (PDB). In contrast, structural data on F-form actin has been more challenging to obtain. During the last years, advances in single-particle electron cryo-microscopy (cryo-EM) have enabled determining high-resolution structures of F-actin from different organisms, such as skeletal muscle α-actin from *Mus musculus* [13], *Oryctolagus cuniculus* [14], *Gallus gallus* [15–17], *Leishmania major* [18] and actin I from *P. falciparum* [19] to resolutions between 2.15–3 Å. The short length and instability of *P. falciparum* actin I has allowed structure determination only of filaments stabilized by jasplakinolide (JAS).

Here, we explored the function of *Plasmodium berghei* actin II in the parasite. We provide a detailed analysis of expression of actin II in gametocytes and zygotes, revealing that the protein is mainly found associated with the nucleus in these stages. Furthermore, mutants of actin II reveal that the protein tolerates fewer changes for its function in the development of oocysts than for the function in gametogenesis. The critical role of the conserved residue H73 was revealed in mutant parasites. In addition, we present near-atomic-resolution cryo-EM structures of F-actin II in the ADP-Mg state with and without JAS and describe the polymerization properties of the so far uncharacterized actin II.

## Results

### Expression of actin II is restricted to the sexual stages

In order to understand in more detail the spatial and temporal localization of actin II in the parasite, we generated transgenic *P. berghei* parasites expressing actin II N-terminally fused to FLAG; the small FLAG peptide was not expected to interfere with the function of the protein. Two different transgenic lines were used in the subsequent experiments, one generated by the conventional approach, while the other was a marker-free mutant produced using CRISPR/Cas9 methodology [20]. Both mutants developed normally at all stages [20].

To determine the timing of expression of actin II, we carried out Western blot analysis of *flag*::*actII* extracts of purified samples from mixed blood stages, activated gametocytes, and zygotes. The blot was probed with a commercial antibody recognizing FLAG to specifically highlight actin II (**Fig 1A**). This revealed that actin II expression was highly upregulated after induction of gametogenesis. The protein was weakly expressed in mature female gametes, and its amount increased in zygotes. These results are consistent with our genetic analysis, which showed that male gametogenesis was abolished in a mutant lacking actin II [10], while on the other hand, maternal expression of actin II in the zygote stage is strictly required for development of the oocyst stages [21]. Extracts of dissected mosquito midguts containing oocysts (harvested 3 and 10 d post-blood feeding) were also analyzed by Western blot; in this case, no signal was detected (**S2A Fig**). These results are consistent with previous results on the transcription of the gene [21].

Immunolabeling using the anti-FLAG antibody to detect actin II was carried out in different stages of the parasite (**Fig 1B–1E**). Male gametocytes showed a dynamic actin II signal. In non-activated gametocytes, the protein was seen mainly in the nucleus (**Fig 1B and 1C**), although a weaker signal was also detected in the cytoplasm (white arrow in (**Fig 1B**), 0 min sample). Later (4 min post activation), its expression was restricted to the nucleus, as revealed by the overlap with DNA. 8 min after activation, the protein was localized in the periphery of the nucleus as long rods in some cells or as dots or very short rods in others. This pattern was seen in a minority of the cells observed (~10%), suggesting that the protein is only briefly present in the rod-like structures. In exflagellating males, the protein was localized in the residual cell with no signal in the flagellar male gamete. In female gametocytes, actin II was not detected (**S2B Fig**), consistent with the results from the Western blot and our previous results [10,21]. Localization of FLAG::actinII was also investigated in zygotes (**Fig 1D and 1E**). In zygotes, two patterns were seen. Most samples showed diffuse actin II in the nucleus. In rare samples, the protein was localized in rods associated with the nucleus. This may suggest that transient filaments are formed during zygote maturation. In ookinetes, no signal was detected (**S2C Fig**), consistent with the Western blot analysis.

These results are consistent with our previous studies of mutants lacking actin II, but also offer new insights into the function of the protein. The timing of expression coincides with the stages that are affected in the mutants. The cytoplasmic localization in non-activated

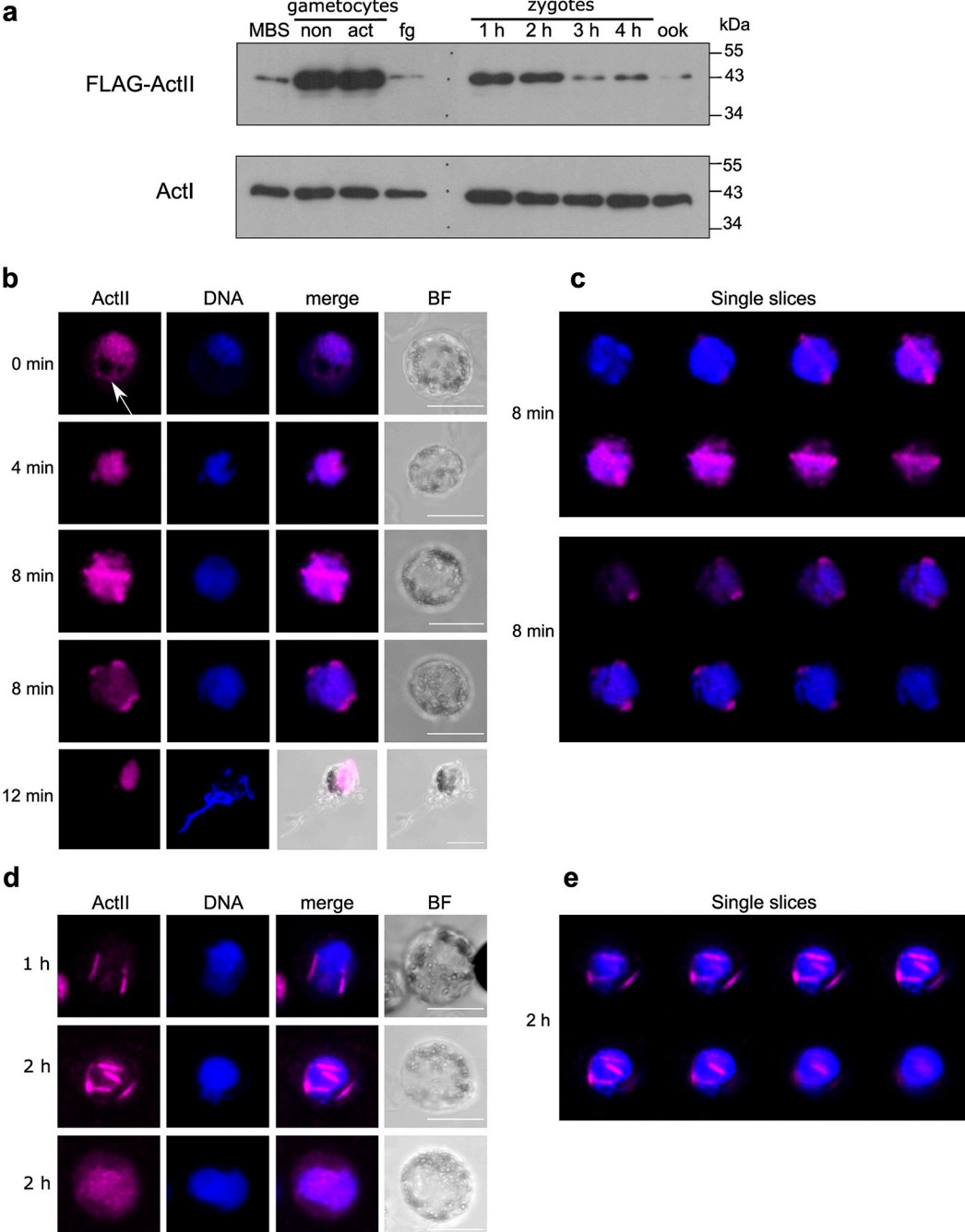

**Fig 1. Expression of actin II in mosquito stages. (a)** Western blot of crude extracts of gametocytes, zygotes, and ookinetes. The lanes from left to right contain: mixed blood stages containing asexual parasites and gametocytes (MBS); non-activated (non) gametocytes; activated (act) gametocytes; female gametes (fg); zygotes 1, 2, 3, and 4 h post activation; ookinetes (ook). mw denotes the size (in kDa) of the molecular weight markers. Actin II was visualized with an anti-FLAG antibody. For loading controls, a duplicate blot was probed with a monoclonal antibody specifically recognizing actin I [22]. The exposure time was different for the two blots, as actin I is expressed at much higher levels than actin II. **(b, c)** Localization of actin II in male gametocytes. Samples of infected blood were fixed at different times (indicated on the left) after activation by dilution in ookinete medium. Cells were labeled with a commercial anti-FLAG antibody, followed by a conjugated secondary antibody (Alexa-647; pink), and DNA was stained with Hoechst 33342 (blue). Controls with only the secondary antibody were included in the experiment; these had no signal (not shown). Imaging was done on a confocal microscope, and the images are projections of z-stacks. Scalebars: 5 μm. **(c)** Montage of single slices of the z-stacks of the two 8 min samples shown in (b). **(d, e)** Localization of actin II in zygotes. Zygotes were purified at 1 and 2 h after activation (indicated on the left) by magnetic beads carrying an antibody directed against the P28 surface protein, which is expressed in activated female

gametes and zygotes. Cells were labeled as in (b). Controls with only the secondary antibody were included in the experiment; these had no signal (not shown). Imaging was done on a confocal microscope, and the images are projections of z-stacks. Scalebars: 5 μm. **(e)** Montage of single slices of the z-stacks of the 2 h sample shown in (d) (middle row).

gametocytes suggests a direct role of actin II in the release from the surrounding host cell. However, the association with the nucleus in activated gametocytes and zygotes was unexpected. Furthermore, the rod-like structures detected suggest that filaments are transiently formed.

## Near-atomic-resolution structures of filamentous actin II

Actins are very conserved among eukaryotic organisms, owing to their essential functions in different cellular processes. Significant differences in terms of sequence conservation of actin II compared to actin I and α-actins involve the D-loop and subdomains (SD) 3 and 4 (**S3 Fig and S1 Movie**). To understand the molecular basis of the filament-forming properties of actin II required specifically in the sexual stages of the parasite life cycle, we set out to determine a high-resolution structure of recombinantly produced actin II in filamentous form in the absence and presence of JAS, which has previously facilitated structure determination of the highly unstable actin I filaments.

The average resolution of the actin II filaments with and without JAS are similar, 3.3 and 3.5 Å, respectively, according to Fourier correlation, using the 0.143 criterion (**S1 Table and S4 Fig**). The actin core is well resolved in both structures with resolutions similar to other filamentous actins [12,13,23,24]. However, the D-loop shows a lower local resolution in both reconstructions (**S5 Fig**). The first four residues of the N terminus were not resolved. We modeled six subunits of the filament into the density in both cases. Density for ADP and $Mg^{2+}$ was clearly identified in both maps (**Fig 2**).

## Conformation of protomers in filamentous parasite actins

The helical twist angle for filamentous actin II was -167˚ and was not affected by JAS. Thus, the helical parameters of actin II are closer to α-actin (-167˚) than to actin I (-168˚), as we have also observed before in low-resolution reconstructions [5,24,25]. The overall conformation of the filament and the SDs in the protomers resemble those of other actins. As the filament grows, the D-loop (SD2) at the pointed end inserts into the C terminus (SD3) at the barbed end of the next protomer. The nucleotide-binding pocket located between the inner and outer domains (ID and OD) is occupied by ADP and $Mg^{2+}$. As in other actins, the transition from G- to F-form involves significant conformational changes [11,26]. The interdomain dihedral angle (θ) is larger in G-actins and the OD SDs undergoes a "flattening" type of movement upon insertion to the filament (**S2 Movie**). F-actin II has a flat conformation like F-α-actin [13,23,25,27,28] and F-actin I [24]. However, there are slight differences in the θ angles of the different actins. The θ angle of F-actin II with JAS is rotated 0.3˚ relative to the F-actin II, 0.9˚ relative to *G. gallus* F-actin (6DJO), 2.5˚ and 3.5˚ relative to the F-actin I structures (5OGW, 6TU4) (**S3 Movie**). In conclusion, F-actin I is more flattened than the other F-actins, including F-actin II (**S6 Fig**).

Likewise, different conformations of certain side chains in F-actin II were observed compared to G-actin II. The S14 side chain is turned towards H73, which is methylated, and R270 to D184 in F-actin II (**S3 Movie**). Moreover, the C terminus is ordered, except for F376, in which the density of the phenyl group is weak. In both structures, H372 interacts with E117 (3 Å). However, the distance between H372 and K113 is longer in F-actin II structures (5.6 Å and 5.5 Å) than F-actin I or G-actin II (4.3 Å and 5 Å) (**S7 Fig**). In contrast, the distance between

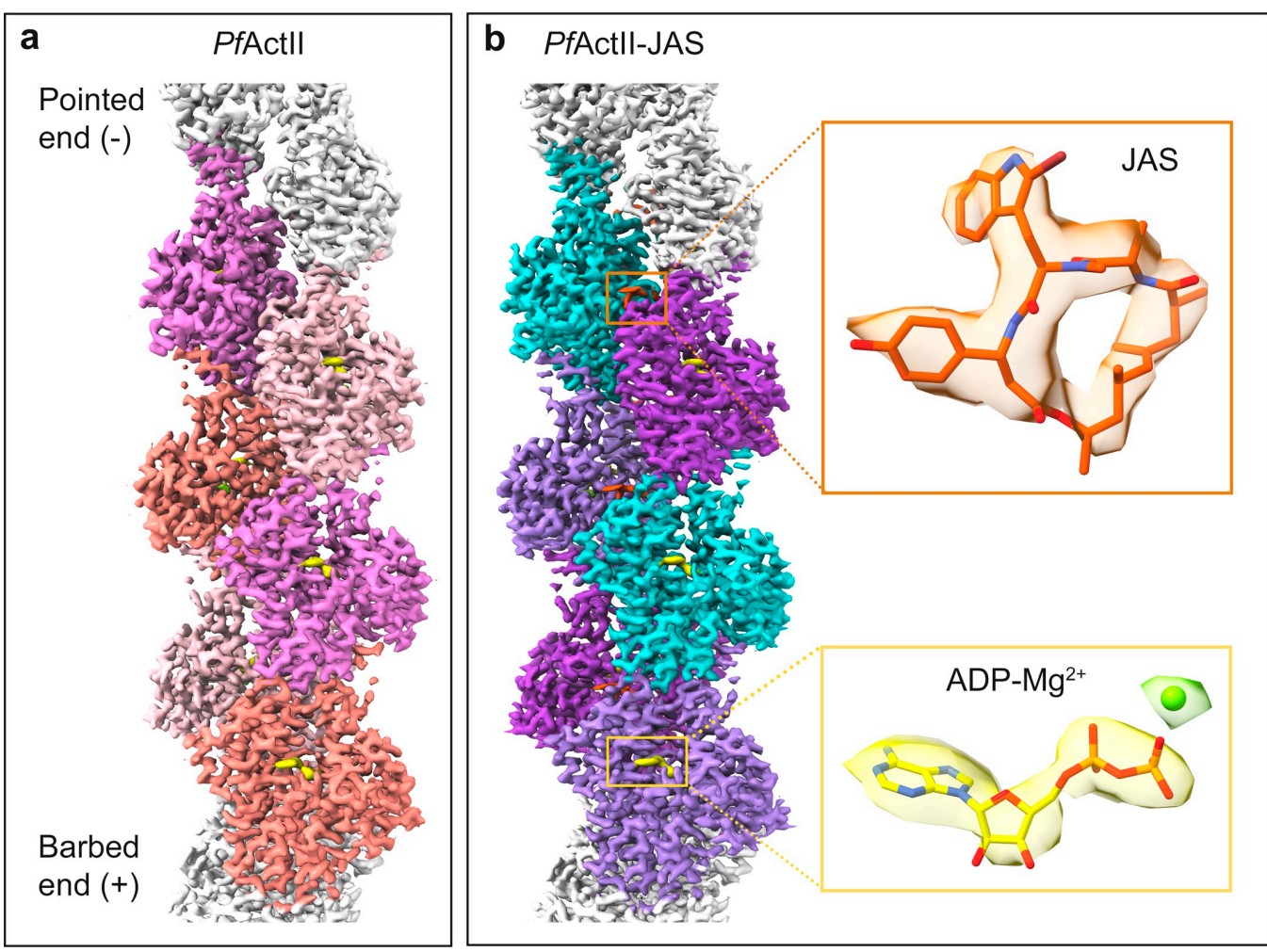

**Fig 2.** Structure of filamentous actin II. Reconstructions of actin II **(a)** and JAS-stabilized actin II **(b)** filaments. Six actin protomers were built and are shown in the density map in different colors. Density for ADP-Mg$^{2+}$ is shown in yellow and for JAS in orange.

H372 and K114 in actin I is ~3 Å because of the displacement of the N-terminal tip of the helix [29].

## Protomer interactions within the filament

In filamentous form, individual actin protomers interact with protomers from the same strand (intrastrand interactions) and the opposite strand (interstrand interactions). SD2 and SD4 at the pointed end of one protomer interact with SD3 at the barbed end of the adjacent protomer of the same strand. The D-loop (39–50), W-loop (165–172), and some regions of SD3 (270–274, 283–294) and SD4 (195–208, 243–247) are involved in the intrastrand interactions (**S8 Fig**). In skeletal muscle α-actin, the primary intrastrand contact is caused by a hydrophobic key (I287 of SD3) and lock (a groove of SD4) interaction. The residues S199 and V201 in α-actin are replaced by T199 and T201 in actin II and G200 and S202 in actin I. These changes decrease the hydrophobicity around the key-lock contact. The interaction is even weaker in F-actin I, caused by the replacement of I287 (in α-actin) by a V288 [24] (**S9 Fig**).

Another intrastrand interaction is mediated by the D-loop in SD2 (**Fig 3**); the D-loop interacts with the adjacent protomer *via* hydrophobic interactions and is indispensable for

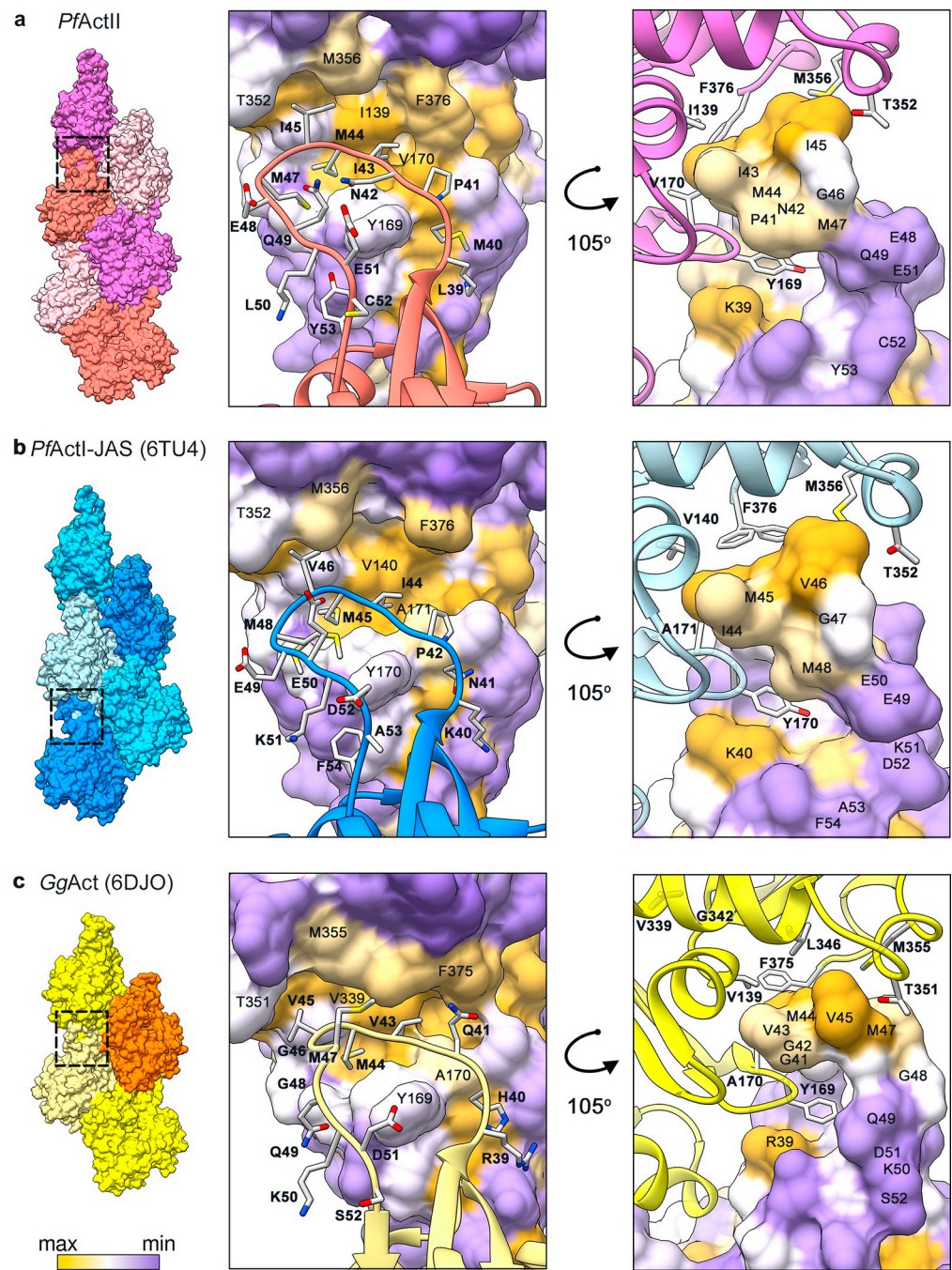

**Fig 3. Interaction of the D-loop with the adjacent protomer. (a)** *P. falciparum* actin II, **(b)** actin I-JAS (6TU4), **(c)** skeletal muscle α-actin from *G. gallus* (6DJO). The actin surface is colored according to hydrophobicity; high (yellow), medium (white), and low (purple).

polymerization [30]. In most crystal structures, the D-loop is disordered. The D-loop in *Plasmodium* actins differs from canonical actins by the substitution of four residues, R39, G41, V43, and G48 (in α-actins; **Fig 3C**) by K40, P42, I44, and E49 (in actin I; **Fig 3B**) or K39, P41, I43, and E48 (in actin II; **Fig 3A**). The substitution of G48 (non-polar, neutral, no side chain) by E49 in actin I or E48 actin II (polar, negatively charged, large side chain) is the most

significant change. In skeletal muscle α-actins, Y169 forms an N-H⋯π bond with the side chain of Q49 (**Fig 3C**), contributing to filament stability [25]. In *Plasmodium* actins, the side chain of either E50 (F-actin I) or Q49 (F-actin II) is in an opposite orientation (**Fig 3A and 3B**). Interestingly, the substitution of H40, V139, and A170 (canonical actins) by M40, I139, and V170 in actin II increases the hydrophobicity in actin II (**Fig 3**). Only one substitution exists between actin I and actin II, G43 in actin I, by a polar amino acid N42 in actin II (**Fig 3A and 3B**).

The interstrand contacts involve principally electrostatic interactions between the plug region (SD3) and the SD2 of a protomer from the opposite strand (**Fig 4**). This interaction is structurally conserved. However, in *Plasmodium* actins, the substitution of M269 in canonical actins (**Fig 4D**) by K270 (actin I; **Fig 4C**) or R270 (actin II; **Fig 4A and 4B**) inverts the electrostatic potential. Previous studies on *T. gondii* and *P. falciparum* actins showed that mutation of K270 to M increases the filament stability [29,31]. In actin II, R270 does not seem to cause shortening of the filaments. K270 in actin I increases the positive charge of the surface more than R270 in actin II (**Fig 4A–4C**). This could be a contributing factor to the actin II filaments being more stable than F-actin I. In addition to the electrostatic interaction of the plug, there are other residues involved in the interstrand contacts; H173 (SD3) and L113 (SD1) of one protomer with L39 (SD2) and the residues E195 and G269 (SD4) of the opposite protomer (**S10 Fig**).

## Long and stable actin filaments alone are not sufficient for oocyst development

Previously, we analyzed a chimeric actin where the D-loop of actin I was replaced with that of α-actin. *In vitro*, this chimeric actin, expressed in a heterologous system, formed long and stable filaments. We also generated a mutant expressing the chimera from the *actin II* locus in the *actII(-)* background. This mutant showed normal exflagellation [5]. Here, we tested whether this mutant could form normal oocysts. Mosquitoes were fed on a mouse infected with this chimeric (*actIchi*) mutant, and midguts were dissected and labeled with an antibody against the oocyst capsule protein Cap380. This revealed that oocysts were formed, although in significantly reduced numbers compared to the WT. They remained small, and no sporozoites developed, thus resulting in a block in mosquito transmission (**Fig 5**). Thus, for normal oocyst formation, other specific features of actin II seem to be necessary, not simply the ability to form long and stable filaments.

## Histidine 73 is crucial for actin II function in oocyst development

From our earlier high-resolution structures [5], we saw that unlike H74 in actin I, around 90% of actin II has a H73 methylation when expressed in insect cells (**S11A and S12 Figs**). Using matrix-assisted laser desorption ionisation time-of-flight mass spectrometry (MALDI-TOF MS), we confirmed that H73 is methylated in a significant fraction of actin II (**S11B Fig**), as is the case also in most canonical actins. Methylation of this histidine stabilizes the filament [32]. In actin I, the so-called A-loop plays a role in defining filament length. In the proposed model, the A-loop undergoes a ping-pong movement between two positive charges, such that D180⁻ interacts either with K270⁺ or the unmethylated, protonated H74⁺, promoting stable and unstable filament conformations, respectively [29]. In the case of F-actin II, the A-loop has a conformation similar to canonical actins and the G-actin II structure, where D179 is oriented towards R177 (**S13 Fig**) and does not interact with the methylated H73. Thus, neither in the crystal structures of actin II [29] nor in the filament structures described here, is the "unstable" conformation detected.

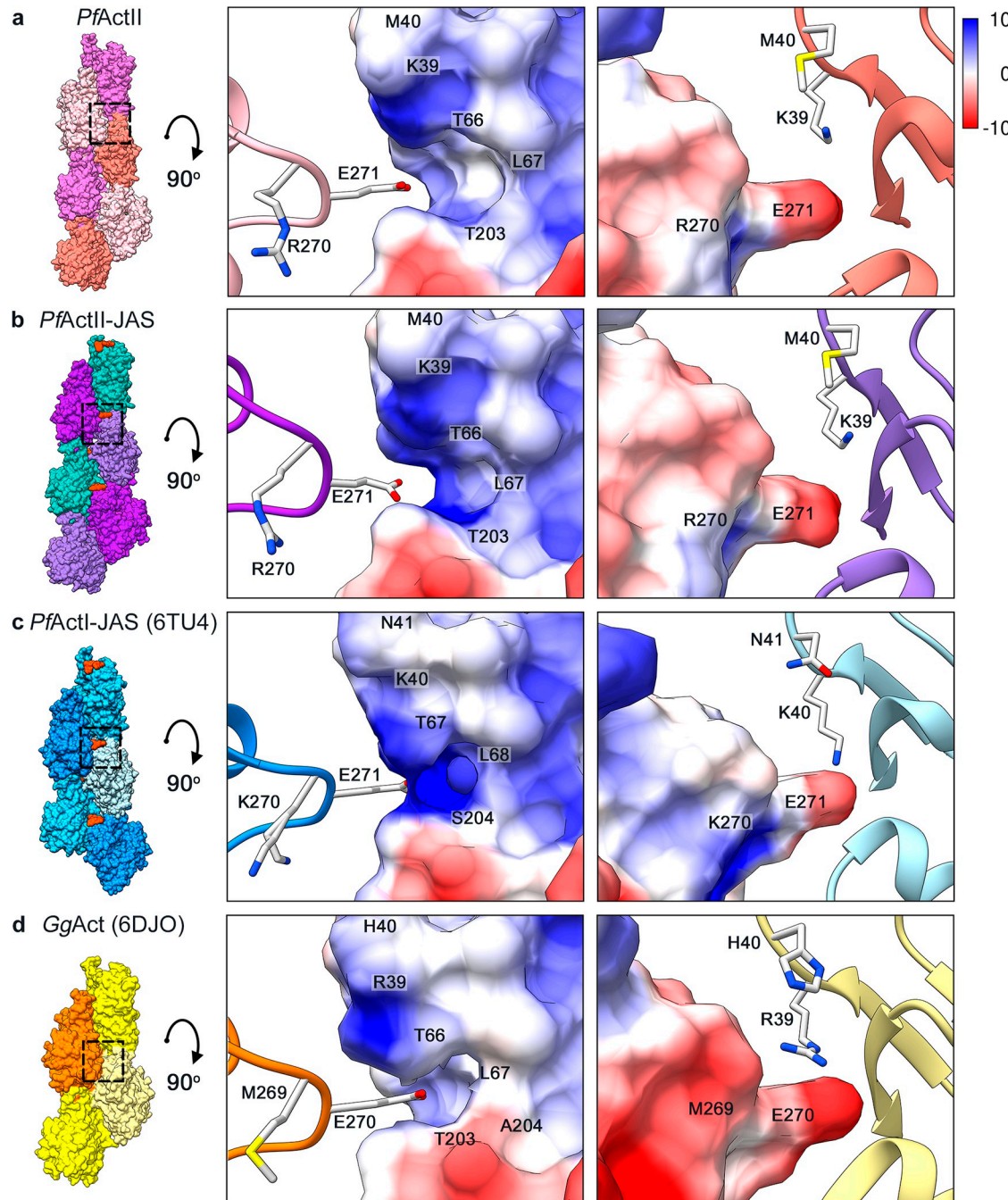

**Fig 4. Electrostatic interactions between the plug and the D-loop.** (a) *P. falciparum* actin II, (b) actin II-JAS, (c) actin I-JAS (6TU4), (d) *G. gallus* skeletal muscle α-actin (6DJO). The surface colored by electrostatic Coulomb potential ranges from −10 kcal·mol⁻¹ (red) to +10 kcal·mol⁻¹ (blue), and the opposing subunits are displayed as ribbon.

To abolish the regulation by methylation, we introduced a point mutation in the *actin II* gene, obtaining the *actIIH73Q* mutant (**S14 Fig**). This mutant was able to normally form male gametes, as exflagellation took place and ookinete conversion was comparable to the WT (**Fig 6A and 6B**). Thus, a histidine at position 73 and its possible methylation are not important for the function of actin II in these early mosquito stages. Next, mice infected with WT and the

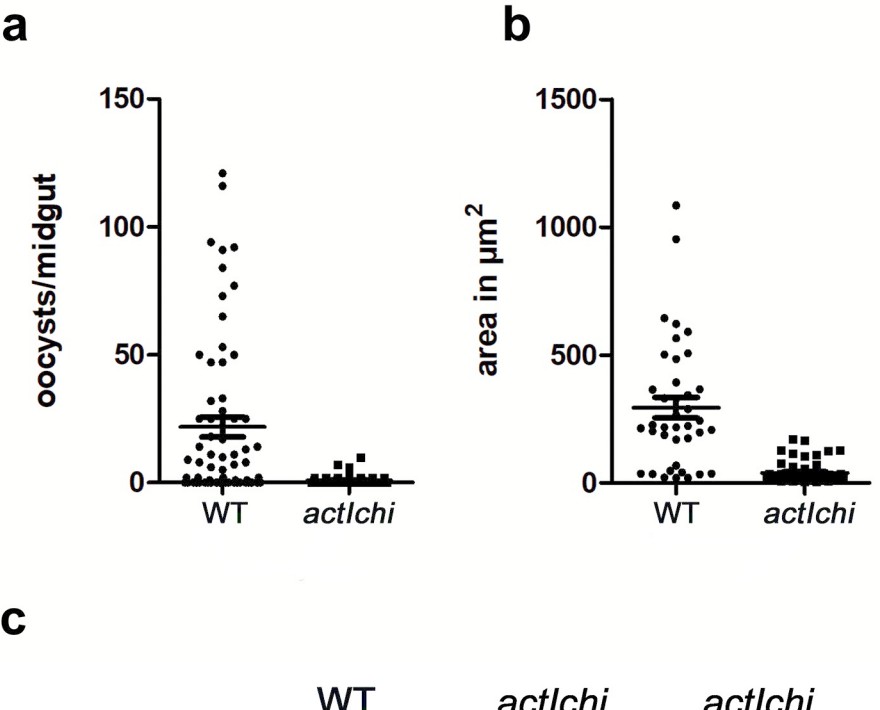

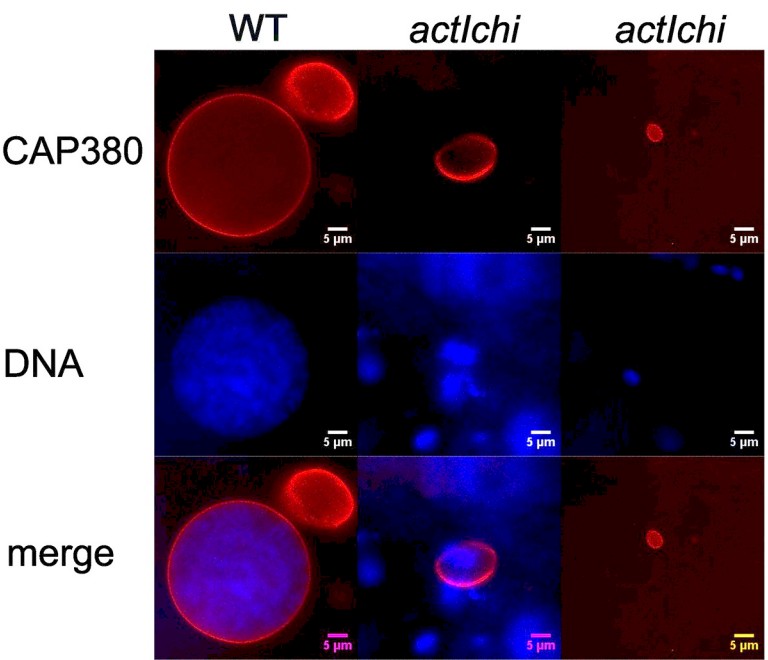

**Fig 5. Oocyst development is blocked in the *actIchi* mutant.** Oocysts at day 12 in mosquitoes fed on WT and *actIchi* infected mice. **(a)** Number of Cap380 labeled oocysts per midgut (WT n = 39, *actIchi* n = 61). Difference is significant, Mann-Whitney test, P<0.0001. **(b)** Area in $\mu m^2$ of WT and *actIchi* oocysts. The area of 39 WT and 61 mutant oocysts from the experiment in **(a)** were measured in ImageJ. Differences are statistically significant p<0.0001 Student's t-test. **(c)** Examples of WT and two representative types of mutant oocysts labeled with the Cap380 antibody (red) and stained for DNA with Hoechst 33342 (blue). Scale bars 5 μm. Raw data related to panels a and b can be found in S1 Data.

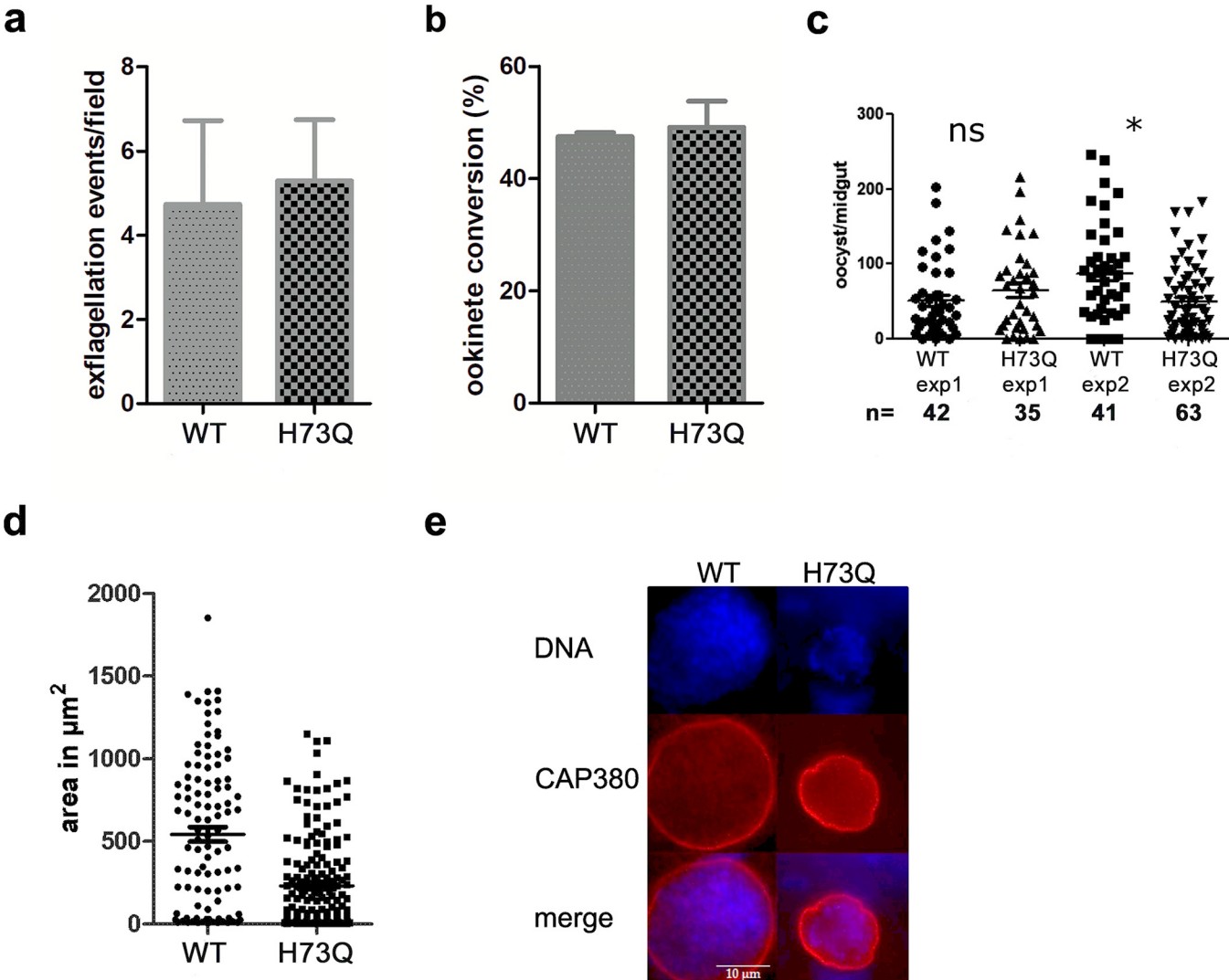

**Fig 6. Methylation of actin II is required for oocyst development. (a)** Exflagellation of *actIIH73Q* was comparable to WT. Exflagellation tests were carried out three times for the WT and eleven times for the mutant. Error bar denotes SEM. Differences are not significant (Student's t-test). **(b)** Ookinete conversion of the mutant compared to WT determined as the percentage of zygotes that developed into ookinetes. Error bar denotes SEM. Differences are not significant (Student's t-test). **(c)** Number of oocysts/midguts from mosquitoes fed with WT or the *actIIH73Q* mutant. Two independent experiments are shown, number of midguts (n) in each experiment are shown below the graph. The number of oocysts was determined after staining the dissected midguts with the antibody against the Cap380 protein of the oocysts. All labeled oocysts were counted irrespective of size. There was no significant difference comparing WT to mutant, Mann-Whitney test. **(d)** Area in μm² of WT and *actIIH73Q* oocysts. The area of 106 WT and 180 mutant oocysts from the experiment in **(c)** were measured in Image J. Differences are statistically significant p<0.0001 Student's t-test. **(e)** Representative oocysts of WT and *actIIH73Q* mutant. Oocysts were labeled with Cap380 and DNA stained with Hoechst 33342. Raw data related to panels a-d can be found in S2 Data.

mutant were offered to *A. gambiae* female mosquitoes. After 12–13 days, midguts were dissected and labeled for the oocyst capsule protein Cap380, and DNA was stained. All oocysts were counted, even those that were small, and this revealed only a minor decrease in the number of oocysts in the *actIIH73Q* strain comparing to the WT (**Fig 6C**). However, the mutant oocysts were significantly smaller than the WT (**Fig 6D and 6E**). In addition, 80% of WT oocysts contained DNA, while only 42% of the mutant oocysts had visible DNA staining. In the mutant oocysts, no sporozoites were formed. To investigate this further, we allowed infected mosquitoes to feed on naïve mice. Giemsa-stained blood samples were observed from day five onwards. This revealed that, while in the WT strain infection was detectable at day five

(two mice), we never detected any parasites in the mice bitten by the mosquitoes infected with *actIIH73Q* (five mice), even ten days after the mosquito feeding. Taken together, this shows that H73 and possibly its methylation in actin II are important for oocyst development, and that the mutant parasites were unable to form infectious sporozoites. This is similar to what we found for the a*ctIchi* mutant, although in that case, the oocyst phenotype was more severe. To confirm that H73 is indeed methylated *in vivo*, mass spectrometry was carried out on extracts from zygotes and this revealed a signal consistent with methylation (**S11 Fig**).

## The state of polymerized actin II at equilibrium

As the ability of actin II to form long filaments is important for development of the parasite in the mosquito host, we wanted to compare its polymerization propensity *in vitro* to that of the better characterized actin I. Actin I in polymerizing conditions is present mostly as monomers, dimers, and oligomers up to around 17–18 units [33]. To shed light on the situation for actin II, we performed sedimentation assays, applying high relative centrifugal forces of 100 000 and 435 000 *g*. At the highest speed, the amount of actin II in the pellet was comparable to actin I (73% *vs.* 69%, p = 0.38, Student's t-test). The amount of actin II-JAS in the pellet after centrifugation at 435000 *g* increased by approximately 9% compared to actin II without JAS (p = 0.001, Student's t-test) (**Fig 7A and 7B**). Similarly to actin I, the supernatant fraction of actin II could be re-pelleted (75%) 16 h after the initial centrifugation step (**S15 Fig**).

We employed dynamic light scattering (DLS) to study the size distribution of the actin II samples. In non-polymerizing conditions, actin II appeared to be in a mostly dimeric form with a hydrodynamic radius ($r_H$) of 5.1 nm in comparison with the G-α-actin (2.7 nm) previously reported [33]. Under polymerizing conditions, actin II was distributed into three populations with $r_H$ of 8.1, 43, and 410 nm, with volume percentage contributions of 51, 33, and 17%. Conversely, actin I in polymerizing conditions was more heterogeneous, consisting of dimers with an $r_H$ of 5.4 nm (92%) and a heterogeneous population with an $r_H$ between 11 and 39 nm (8.33%). JAS-stabilized actin II was distributed into four populations: 16, 69, 210, and 680 nm, comprising 47%, 16%, 1%, and 33% of the total volume, respectively (**Fig 7C and 7D**). Interestingly, we observed some long actin II filaments of $\geq$ 1.5 μm using negative-stain electron microscopy, similar to the maximum length of F-actin I reported previously [29]. The remaining protein was present as monomers, dimers, or trimers. In contrast to actin I, shorter filaments or filament-like structures between 0.3–1.5 μm were not observed (**Fig 7E**), despite the 43 nm peak present in DLS (**Fig 7C**).

In conclusion, our data show that, under polymerizing conditions, at the steady-state, actin II is distributed into three populations comprising roughly 4, 18, and $\geq$180 protomers. In non-polymerizing conditions, the major population corresponds to a dimer. Overall, the F-actin II oligomers and polymers appear larger than those of actin I, which is mainly present as dimers and filament-like structures from 4 to 17 protomers. JAS increases the pelletable actin II, especially at the lower g forces, due to its stabilizing effect.

## Jasplakinolide does not affect the overall actin II filament conformation

JAS stabilizes both actin I and II as well as canonical actin filaments. However, unlike for actin I, JAS is not required for obtaining stable long filaments of actin II. Furthermore, JAS has been postulated to induce changes in the filament conformation of α-actin [34]. Therefore, we wanted to confirm whether JAS affects the filament conformation for actin II.

In α-actin, JAS promotes conformational changes in the D-loop (open state) and the C-terminus (disorder) [23]. The D-loop in our structures has a closed conformation (P41 and E48) (**S16 Fig**). Only the residue located in the filament interface is substituted by Y199 while Y53

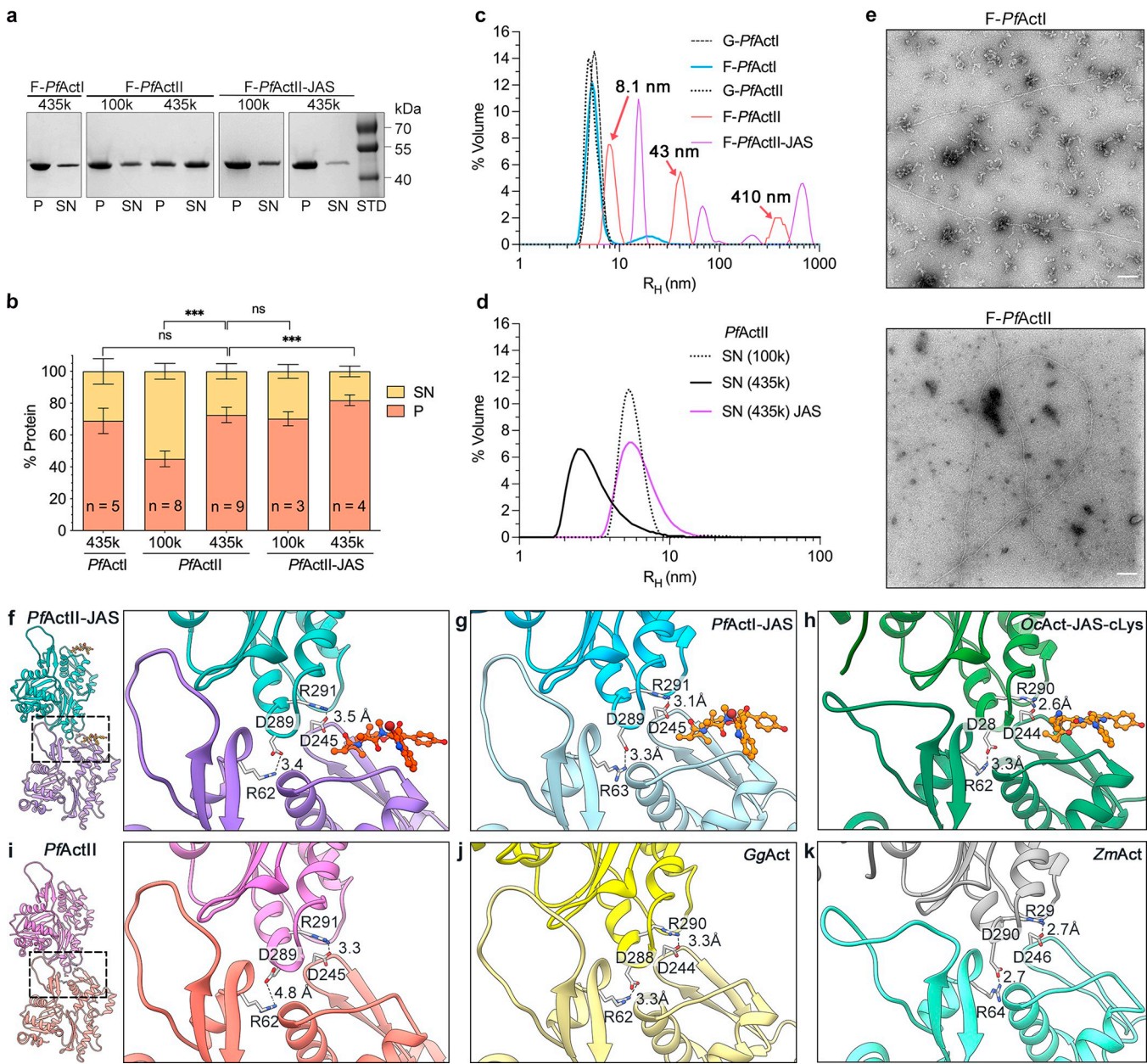

**Fig 7. Polymerized *Plasmodium* actins. (a).** SDS-PAGE of a standard sedimentation assay of actin I and actin II, where the pellet (P), and supernatant (SN) fractions have been separated by ultracentrifugation at 4°C for 1 h. **(b).** Quantification of the sedimentation assays expressed as the percentage of protein in each fraction after ultracentrifugation at 100000 or 435000 *g*. The samples were polymerized in the presence or absence of JAS in a ratio of 1:1. **(c-d)** DLS profile of *Plasmodium* actins.The 8.1, 43, and 410 nm peaks in the F-*Pf*ActII sample (panel c, red curve) are indicated with arrows. **(e).** EM micrographs of polymerized Actin I (1 μM) and actin II (2 μM). Scale bars represent 100 nm. **(f-k)** Salt bridge interactions between two adjacent protomers. Actin filaments in the presence of JAS **(f)** actin II, **(g)** actin I (6TU4), **(h)** α-actin (5OOC). Structures without JAS **(i)** actin II, **(j)** α-actin (6DJO), **(k)** *Z. mays* actin (6IUG). Distances are indicated with black dashed lines. The error bars represent the standard deviation, \*\*\*<0.001, \*<0.05, and ns: not significant, two-tailed Student's t-test. Raw data related to panel b can be found in S3 Data and panels c and d in S4 Data.

remains the same as in α-actin. Furthermore, JAS reduces the $P_i$ release rate without modifying the polymerization rate [35,36]. In JAS-stabilized actin II, density for the leaving phosphate is not observed.

Close to the D-loop, the residues R62 (SD2) and D245 (SD3) form two important salt bridges with D289 and R291 (SD3) of the adjacent protomer. Notably, the distance between R62

and D289 in JAS-stabilized F-actin II is shorter than in F-actin II without JAS. A similar effect occurs in α-actins with the residues R290 and D244. Interestingly, *Zea mays* F-actin was suggested to be more stable than α-actin filaments and JAS-stabilized actin I. Without JAS, *Z. mays* F-actin exhibits an open D-loop conformation, and the distances of the salt bridge interactions are shorter than in α-actin [37] (**Fig 7F–7K**). These observations suggest that multiple intrastrand interactions work synergically to contribute to the stability of the actin filament. Furthermore, JAS does not affect the overall architecture of the actin II filaments or the conformations of the D-loop and the C terminus. It is plausible that this is the case for both *Plasmodium* actins.

## Actin II has a critical concentration similar to actin I and canonical actins

Previously, the critical concentration ($C_c$) of actin I has been reported to be 0.1 μM [33] or ~4 μM [38], depending on the method used, the time of polymerization, and what length is considered as polymerized. To assess whether the $C_c$ of actin I and II, determined using pyrene-actin fluorescence, is dependent on the polymerization time, *Plasmodium* actins were labeled with pyrene [33] and polymerized for 1 or 20 h at 4°C. Concentrations above 5 μM had not been tested before for actin I using this method [33].

The $C_c$ was calculated as the first concentration point, where a linear increase in fluorescence signal started. For both actin I and II, we obtained a $C_c$ identical to the one observed previously for actin (0.1–0.2 μM) after polymerizing for 1 h (**Fig 8A–8D**). Interestingly, a second linear slope was observed after 20 h of polymerization for actin I and already 1 h after for actin II. This second intercepting point gave Cc's of 1.9 μM for actin I and 4.6 or 3.0 μM for actin II after 1 and 20 h, respectively. A linear regression curve was also observed for actin II polymerized in the presence of JAS (ratio 1:1). The $C_c$ did not change significantly upon addition of JAS either after 1 or 20 h of polymerization (**S17 Fig**). Thus, JAS seems to merely affect the stability and not the $C_c$ of the *Plasmodium* actin filaments.

Based on these data, we conclude that both *Plasmodium* actins exhibit a $C_c$ similar to that of canonical actins. Besides, the $C_c$ plot at the steady-state is not linear, and the two slopes that we observed could be related to different populations of actin II filaments. We speculate that the first slope corresponds to the excess of nucleation at concentrations below 3 μM, and the second slope could be the elongation phase of longer filaments.

## Polymerization and depolymerization kinetics of actin II

We wanted to compare the polymerization kinetics of the two *Plasmodium* actins. To keep them monomeric, the proteins were purified in 0.3 M ammonium acetate, which was removed immediately before performing the polymerization assays. Although a lag phase was observed for both actins, it was less pronounced for actin II. Actin II also reached equilibrium faster than actin I. After 100 min, concentrations below 5 μM were in equilibrium for actin II, while for actin I, the equilibrium was reached in approximately 120 min at 5 μM (**Fig 8E and 8F**). This may reflect the instability of actin I filaments. The actin II polymerization curves exhibited an overshoot, which presents itself as a peak in the first 10 min of polymerization both in the presence and absence of seeds. The overshoot was less pronounced at lower actin concentrations and in the presence of actin II nuclei (**Fig 8F and 8G**). This overshoot has also been reported for canonical actins and is related to fast ATP hydrolysis that causes partial depolymerization of the actin-ADP subunits at the pointed end of the filament. This phenomenon can increase by branching, severing by sonication, and the presence of ADF/cofilin or seed filaments [39–41].

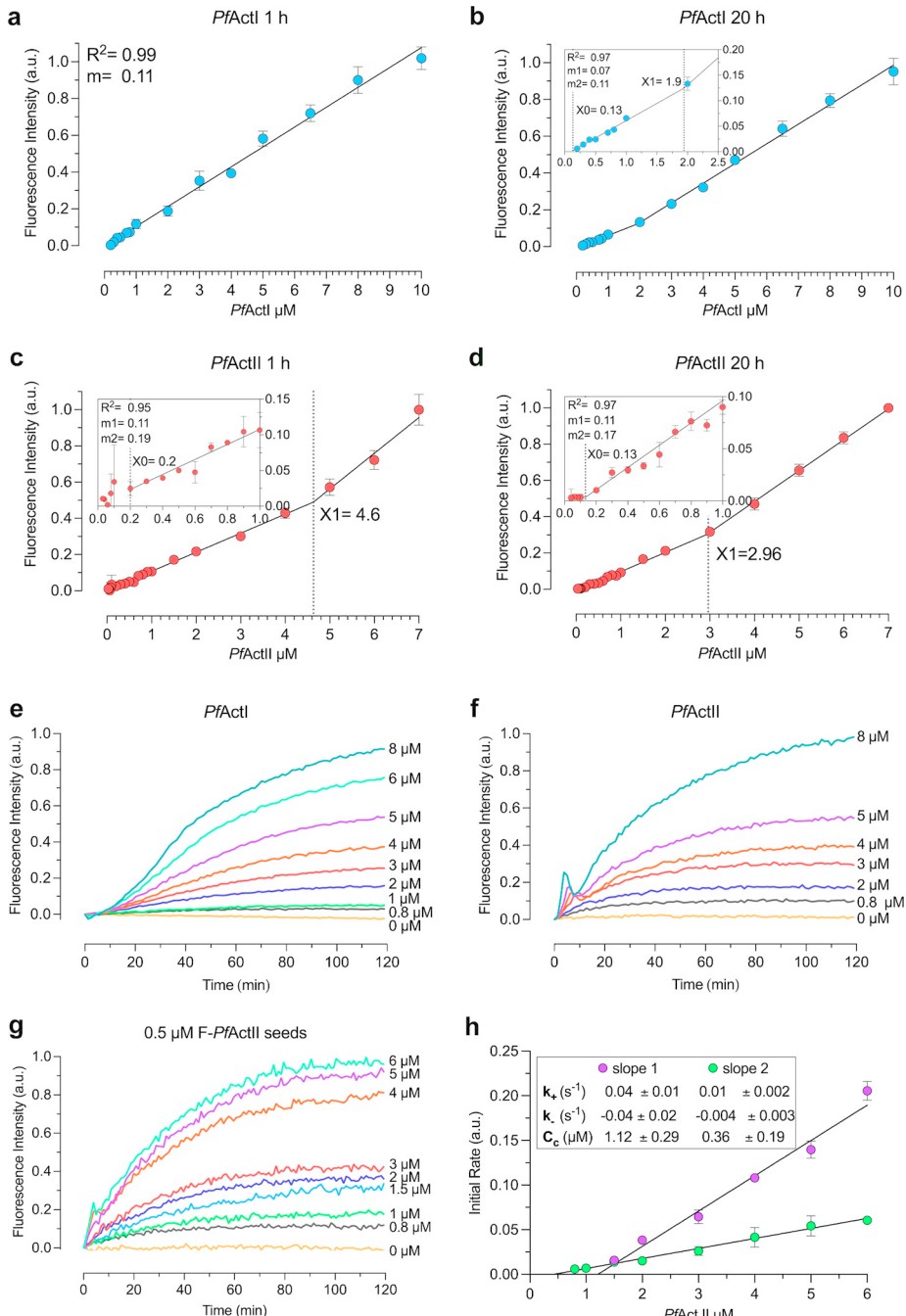

**Fig 8. Polymerization kinetics of *Plasmodium* actins. a-d.** $C_c$ plots of *Plasmodium* actins in F-buffer. $C_c$ plot of actin I after 1 h ($n^b$ = 2) **(a)** and 20 h ($n^b$ = 2) **(b)** of incubation at 4°C, using 0.002 to 10 μM actin. The data fit a single regression line in **(a)** and a segmented equation in **(b)**. X0 agrees with the $C_c$ previously reported (0.13 μM), and X1 was 1.9 μM. $C_c$ plot of actin II after 1 h ($n^b$ = 5) **(c)** and 20 h ($n^b$ = 3) **(d)** of incubation, using 0.002 to 7 μM actin. Actin II exhibited a critical concentration of 0.2 μM (indicated as X0). The $C_c$ decreased to 0.13 μM after 20 h, as seen in **(d).** X1 indicates the crossing point value between the two slopes. The lower concentration points are zoomed on the left. Actin II polymerization **(e-h)**. Spontaneous polymerization curves of pyrene-labeled actin I ($n^t$ = 3) **(e)** and actin II ($n^t$ = 3) **(f)**, induced by the addition of 1x F-buffer. **(g)** Nucleated polymerization curve of actin II ($n^t$ = 3). **(h)** Elongation rate constants of actin II of the initial phase (slope 1) and the second phase (slope 2) ($n^b$ = 3). Error bars represent the standard deviation, a.u. = arbitrary units; $n^t$ = technical and $n^b$ = biological replicates with 3 technical replicates per experiment. Raw data related to panels a-d can be found in S5 Data, panels e and f in S6 Data, and panels g and h in S7 Data.

For determining the elongation rate constant, actin II was polymerized in the presence of 0.5 µM pre-polymerized actin II seeds [33]. The first 20 min of polymerization exhibited two different phases, as was previously observed for actin I using skeletal muscle α-actin seeds [33]. The first phase was 4-fold faster than the second. Using the $C_c$ value obtained from the steady-state and the equation from the second phase, the corresponding relative value of $k_-$ was 0.003 ± 0.003 s$^{-1}$ (**S2 Table**).

In conclusion, actin II polymerization curves exhibited three phases: lag, elongation, and treadmilling phases like other actins. The observed $k_+$ and $k_-$ were lower compared to α-actin and actin I [33]. This could be related to the overshoot which was observed in the first 5–10 min of polymerization.

## Discussion

### Actin II expression and localization

The presence of actin II only in male gametocytes and female gametes and zygotes confirms our earlier analyses of mutants lacking actin II [10,21]. Here, we show for the first time that actin II is associated with the nucleus both in the male gametocyte and in the zygote. Immuno-labelling of samples obtained during the short time of gametocyte development (~10–15 min) revealed a dynamic pattern. In non-activated samples, actin II was mainly in the nucleus, but a cytoplasmic signal in the cell was also visible. This may indicate that it is directly involved in the rupture of the parasitophorous vacuole membrane and the host cell membrane, consistent with the earlier finding that, in the *actin II* deletion mutant, rupture of the membranes was blocked [10].

In developing male gametocytes, mainly a diffuse signal was detected in the nucleus, but in rare occasions of cells fixed at 8 min p.a., actin II was seen as distinct dots or rods associated with the nucleus. This pattern resembles the localization of tubulin [42,43], but attempts to verify co-localization were inconclusive. In the actin II deletion mutant, DNA replication took place but karyokinesis did not. Furthermore, axonemes that were normally formed were not activated. Thus, actin II may be associated with the axonemes that are formed in close association with the nucleus [44]. In zygotes, the protein also associated with the nucleus in two distinct patterns. In most samples, there was a diffuse signal, while in rare cases, rod-like structures were seen. We have previously shown, using a mutant encoding GFP-actin II, that overexpression of actin II blocked parasite development once meiosis was completed. Furthermore, ookinetes derived from a genetic cross between WT male gametes and *actII(-)* females had a tetraploid value of DNA [21].

Taken together, these data argue against a direct role for actin II in meiosis. Actins in higher eukaryotes have important functions in the nucleus, being involved in transcriptional regulation, chromatin organization, mitosis, and meiosis as well as nuclear structure [45]. We can exclude a specific role in transcriptional regulation based on a transcriptomics analysis of gametocytes of the *actin II* deletion mutant [46], but a more exact understanding of the role of actin II must await further studies.

### Differences between *Plasmodium* actins affect the length of the filaments

The most obvious difference between *Plasmodium* actins and canonical actins is the short length of the filaments observed *in vitro* [5,29,47]. The first explanation suggested for this was an isodesmic mechanism of *T. gondii* actin assembly [48], which has subsequently been shown not to be the case for *Plasmodium* actin I [33,38]. Another explanation for the filament length differences is a model based on fragmentation, proposed for actin I. In this model, filament stability is determined by the interactions of D180 in the A-loop with either K270 (plug) or

H74 in the H-loop (depending on its protonation state). This promotes a ping-pong movement of the A-loop, causing a change of the θ angle, which acts as a switch between stable and unstable filament conformations [29].

While spontaneous fragmentation appears to be more favorable for actin I, it does not seem to be the case for actin II. However, the length of the filaments in actin II seems to be shorter, and the occurrence of long filaments lower than in canonical actins [5]. Fragmentation in canonical actins is dependent on filament length [49]. Possibly, it is the case also for actin II since there is no evidence of a ping-pong movement of the A-loop. In comparison with actin I, the fragmentation of actin II could be less favorable due to structural differences that do not permit the second conformation of the A-loop. Instead, fragmentation could be mediated *in vivo* by other mechanisms, such as ABPs or the intracellular environment. Such factors remain to be characterized. Actin II has a cysteine in position 272, like non-muscle β and γ-actins. This cysteine is the most reactive residue towards $H_2O_2$ in solution, and its oxidation causes negative effects on ABP interactions and depolymerization [50]. In actin I, this residue 272 is an alanine, and the mutation A272C favored the stable A-loop conformation [29]. Perhaps, reactive oxygen species could have different effects on the depolymerization and/or fragmentation of the two *Plasmodium* actins *in vivo*.

While lateral contacts between actin monomers are important for filament stability, the longitudinal contacts are also essential. The D-loop plays a major role in longitudinal interactions. The largest differences in the *Plasmodium* actin D-loop compared to α-actin are in the tip of the loop. Both *Plasmodium* actins have a substitution of Q42 and G49 (in α-actins) by P41/42 and E48/49 (actin II/I). In actin I, these residues impact $P_i$ release and are possibly related to the closed conformation of the D-loop. The mutation P42Q that increases the filament stability *in vitro* is lethal *in vivo* [51]. Also, in filamentous α-actins, Y169 forms an N-H···π bond with the Q49 side chain [25]. However, E50 (in F-actin I) or Q49 (in F-actin II) adopt an opposite orientation.

Despite the similarity of the D-loop in actin I and II, the P41 and E48 substitutions do not drastically seem to affect the filament length in actin II. Thus, other factors must also play important roles. Especially residues M40, I139, and V170 in actin II may contribute to strengthening the hydrophobic interactions between SD3 of one monomer and the D-loop of the adjacent monomer (**Fig 4**).

## Different requirements for actin II during male gametogenesis and zygote-to-ookinete development

Complementation of *actin II* deletion mutant parasites with a chimeric actin I with the canonical actin D-loop partially restored the phenotype, as exflagellation of the male gametocytes took place normally [5]. In this mutant, oocysts formed, albeit in lower numbers, but they did not develop and remained very small. This phenotype is similar to what we have found for parasites, in which actin I was expressed from the actin II locus in the *actII(-)* mutant background [21]. In that case, exflagellation was however also severely reduced, but the few ookinetes that developed did form oocysts, which remained small. As the chimeric actin I forms long and stable filaments *in vitro*, these results suggest that while the function in exflagellation can tolerate more radical changes in structure, oocyst formation is very sensitive to perturbations.

Further indication for different requirements for actin II in the two stages was obtained from the H73Q mutant parasites. Methylation of H73 is involved in actin monomer stability and flexibility of the OD and ID [52]. In heterologously expressed actin I, the corresponding H74 is not methylated [29]. In actin II, the methylation of H73 seems to stabilize the interaction with D179, favoring the stable A-loop conformation (**S9 Fig**). In the *actIIH73Q* mutant,

male gametogenesis was not affected, but oocyst development was abnormal, and sporozoites never developed. We have described a similar phenotype in mutant parasites where actin II was expressed from a transgene [53]; while exflagellation was normal, oocyst development was aberrant. Taken together, these experiments indicate that the function of actin II in the zygote has strict requirements both for timing of expression and filament stability/length and that regulation by methylation of H73 may be important for the function of actin II in this life cycle stage.

## Effect of JAS on filamentous actin II

JAS stabilizes *Plasmodium* actin filaments without disturbing the architecture of the polymer. The binding site of JAS is the same as for phalloidin; between three protomers. Studies on canonical actins suggested that the D-loop state could contribute to filament stability. An open D-loop state is reported for rabbit skeletal muscle α-actin in the presence of a modified JAS (JAS-cLys), phalloidin in ADP-$Mg^{2+}$ state, and α-actin-ADP–BeFx state [23,34]. Recently, an open D-loop conformation was also observed in F-actin-ADP-$Mg^{2+}$ from *Z. mays*. *Z. mays* F-actin is more stable and rigid than α-actins and JAS-stabilized actin I [37]. On the other hand, other cryo-EM structures of α-actin bound to JAS or with ADP-$Mg^{2+}$, AMPPNP or ADP-$P_i$ did not show changes in the D-loop state [15,25]. For *Plasmodium* actin II, JAS stabilizes the filaments without affecting the D-loop conformation. JAS strengthens the interaction of R62 (SD2) and D289 (SD3) between two adjacent protomers. This is likely the case for actin I as well, although, so far, there are no filament structures available in the absence of JAS.

JAS inhibits $P_i$ release without modifying the polymerization rate [35,36]. Consequently, it was observed that the phosphate is present in cryo-EM structures of α-actins polymerized in the presence of JAS [15,23,25]. In contrast, density for the phosphate is not observed in JAS-stabilized F-actins II in this study or actin I [24]. Yet, the conformation of S14 and the back-door appear as in canonical actins when the phosphate is not yet released. Thus, further studies on the mechanism of phosphate release from *Plasmodium* actins are required to understand its role in filament dynamics.

## Polymerization mechanism of *Plasmodium* actins

To explain the short length and apparent instability of the apicomplexan actin filaments, an isodesmic polymerization mechanism has been suggested for *T. gondii* actin [48]. This would differ from the classic nucleation-elongation process described for canonical actins, being a non-cooperative process that is not dependent on nucleation. This would exhibit itself as a lack of an apparent $C_c$ for polymerization and the absence of a lag phase in the polymerization curve. However, more recently, a $C_c$ has been reported for actin I in two different studies by us and others, albeit the apparent $C_c$ in these differ substantially from each other [33,38]. The differences are likely due to the different methods used for determining the $C_c$ and, thus, the different length of polymers/oligomers considered as filaments. Secondly, the polymerization curves of *Plasmodium* actins exhibit a lag phase, albeit less pronounced than in α-actin, that disappears in the presence of α-actin seeds.

Our earlier data on actin I and the data reported here on actin II together could support a nucleation-elongation, rather than isodesmic, mechanism for both *Plasmodium* actins. However, not all features of the parasite actin polymerization seem to fit that model either. Alternatively, an anticooperative mechanism, where dimers are energetically more favored than elongation of the polymer [54,55], could explain the excessive nucleation, the lower population of large filaments, and the highly unusual abundance of dimers in *Plasmodium* actins. This is something to consider for future investigations.

## Concluding remarks

The actin II isoform is required at two very specific time points in the complex life cycle of the malaria parasite: during male gametogenesis and zygote formation. Our data here reveal that in both these stages actin II forms transient structures, that may represent filaments associated with the nucleus. This finding is complemented with biochemical and structural studies that reveal that actin II *in vitro* behaves to a large extent similarly to canonical actins and forms long and stable filaments, in contrast to the ubiquitously expressed actin I. The requirement for critically defined specific polymerization characteristics can provide an explanation why in *Plasmodium* a second actin isoform evolved, in contrast to other apicomplexan parasites that only possess one actin.

# Materials and methods

## Ethics statement

All animal work was carried out in full conformity with Greek regulations consisting of the Presidential Decree (160/91) and law (2015/92) which implement the directive 86/609/EEC from the European Union and the European Convention for the protection of vertebrate animals used for experimental and other scientific purposes and the new legislation Presidential Decree 56/2013. The experiments were carried out in a certified animal facility license (EL91-BIOexp-02) and the protocol has been approved by the Foundation for Research and Technology–Hellas (FORTH) Ethics Committee and by the Prefecture of Crete (license number # 93491, 30/04/2018 and # 106323, 29/04/2021).

## Parasite strains and methods

The WT *P. berghei* strain was ANKA 2.34. The deletion mutant *act2*⁻::*mCherry* in which the *actin II* ORF was deleted, has been described described [56] as well as the marker-free strain expressing FLAG::actin II under the *actin II* promoter [20].

Parasites were maintained in 6–10 week old OlaTo mice through serial passages with intraperitoneal injection of infected blood. Two days before injection with infected blood, the animals were treated with 100 µl of phenylhydrazine (25 mg/ml stock solution). Parasitaemia and gametocytemia were determined by counting of Giemsa-stained blood smears.

Gametocytes were obtained from blood of an infected animal. For non-activated samples the blood was immediately added to fixative. Gametocytes were activated after diluting infected blood 1:10 in activation medium (RPMI1640 with L-glutamine, 25 mM HEPES, 2 g/L NaHCO3, 10% foetal bovine serum, 50 µM xanthurenic acid, pH 8.0) and incubating the samples at 19˚C. Exflagellation events were counted under the microscope after activation for 10–15 min.

Cultures of ookinetes was carried out using the method described [57]. Activated females were isolated from ookinete cultures 15 min after seeding using magnetic beads coated with the 13.1 antibody. Zygotes were obtained using the same method at different time points after seeding of the culture.

Ookinete conversion was assessed as previously described [10]. Briefly, ookinetes, female gametes and zygotes were labeled with the 13.1 anti-Pbs21 antibody and at least 100 cells were counted in each experiment.

Mosquitoes used were *Anopheles gambiae* strain G3 [57]. Mosquitoes were fed on anaesthetized infected mice. Infected mosquitoes were kept at 19˚C until dissection for oocyst counts or feeding to naïve C57BL/6 mice.

## Western blot

Parasites were lysed by sonication, and the crude lysates were loaded on a denaturing 12% SDS-PAGE gel. After gel electrophoresis the proteins were transferred to nitrocellulose membrane filters by electroblotting. The membrane filters were incubated with the primary antibodies, followed by secondary antibodies conjugated with horse radish peroxide. The signal was detected using the SuperSignal West Pico solution (Pierce Biotechnology).

## Immunofluorescence assay

Gametocytes were obtained from infected blood samples diluted 1:5 in activation medium (RPMI1640 with L-glutamine, 25 mM HEPES, 2 g/L NaHCO3, 10% foetal bovine serum, 50 μM xanthurenic acid, pH 8.0) supplemented with 50 μM xanthurenic acid; samples were incubated for the indicated time at 19°C. 5 volumes of fixative was then added containing 4% formaldehyde in microtubule stabilizing buffer (MTSB, 10 mM MES, 150 mM NaCl, 5 mM EGTA, 5 mM $Mg_2Cl$, pH 6.9). The cells were spun down on 13 mm round cover slips coated with poly-L lysine at 400 g for 10 min at room temperature (RT), permeabilized with 0.2% Triton X-100 diluted in phosphate-buffered saline (PBS) for 2 min, and washed once in PBS for 5 min. The antibody directed against FLAG was diluted in blocking buffer (BB; 2% bovine serum albumin in PBS) and added to the cells and the samples incubated overnight at 4°C. After washes with PBS the secondary antibody in BB were added, and samples incubated for 1 h at RT, followed by washes in PBS and staining with Hoechst 33342 DNA stain. The samples were mounted in Vectashield and imaged in a Leica SP8 inverted laser scanning confocal microscope.

Zygotes were cultured for 1.5 and 3 h and then enriched with 13.1 beads, after which the samples were processed as above.

Analysis of oocysts was carried out as previously described [21]. Briefly, dissected midguts were immediately placed in fixative (4% formaldehyde (Polysciences), 0.2% saponin (Sigma) in PBS) and incubated on ice for 45–60 min. After washes the primary antibody added and the sample incubated 4°C over night, followed by addition of the secondary antibody and staining of DNA with Hoechst 33342. The guts were mounted and viewed as described above.

## Antibodies

The anti-FLAG antibody monoclonal M2 was purchased from Sigma. The antibodies recognizing Pbs21 (13.1 monoclonal antibody) [58], *P. berghei* actin I [22] and Cap380 [59] have been described.

The secondary antibody used in the immunofluorescence experiments was Alexa-647 anti-rabbit obtained from Thermo Fisher Invitrogen, while in the western blot a secondary HRP conjugated antibody was used (directed against rabbit IgG), obtained from Jackson Immunoresearch.

## *Generation of* flag:actin II

The initial flag::actin II construct was generated using the same strategy as previously described for the *Act2com* and *act2rep* constructs [21]. Briefly, a PCR fragment was generated with a 5'- primer encoding FLAG fused to the coding sequence for the N terminus of actin II; the ATG initiation codon was introduced before the *flag* sequence. The reverse primer corresponded to the C-terminus of the protein. The primers are listed in **S3 Table**. Genomic DNA from WT parasites was used as the template for amplification. The fragment was introduced in the vector pSD141, a derivative of the pL0006 vector [42,60], already containing 2.7 kb of the

*actin II* 5'FR and 728 bp of the 3'- FR, using the restriction sites NheI and NotI. The plasmid was linearized with ClaI before transfection of the *act2⁻::mCherry* parasite strain, which had been recycled to remove the resistance cassette [56]. Parasites were cloned by limiting dilution [61]. Correct integration was verified by PCR genotyping.

## Actin purification

The gene encoding *P. falciparum* actin II (PlasmoDB PF3D7_1412500) was synthesized and codon-optimized for insect cell expression in pFastBac HT A vector by Thermo Scientific. *P. falciparum* actin II was expressed in *Spodoptera frugiperda Sf*9 cells (Invitrogen) at 27˚C, as described before [62]. For purification, a fresh pellet was resuspended in lysis buffer (10 mM CHES pH 8.7, 250 mM NaCl, 5 mM CaCl₂, 5 mM imidazole, 1 mM ATP, 3 mM β-mercaptoethanol, 300 mM ammonium acetate) with 1X SIGMAFAST protease inhibitor cocktail (Sigma). The cells were lysate by sonication. The cell lysate was centrifugate for 60 min at 47,850 *g* at 4˚C. 2 ml of Ni-NTA (Thermo Scientific) beads were equilibrated with 10 column volumes (CV) of lysis buffer. The supernatant was filtered (filter 0.45 mm, Sarstedt), applied to the Ni-NTA column, and incubated for 15 min at 4˚C. The Ni-NTA column was washed 2 times with 25 CV wash buffer 1 (10 mM CHES pH 8.7, 5 mM CaCl₂, 10 mM imidazole, 1 mM ATP, 300 mM ammonium acetate), 10–15 mM imidazole and 250–500 mM NaCl. Then, the column was extensively washed with 50 CV of wash buffer 1, 20 mM imidazole, and 1M NaCl. Finally, the column was washed with 25 CV of elution buffer 2 (10 mM HEPES pH 7.5, 0.5 mM ATP, 0.2 mM CaCl₂, 1mM β-mercaptoethanol and 300 mM ammonium acetate). The protein was eluted with elution buffer (10 mM HEPES pH 7.5, 0.2 mM CaCl₂, 0.5 mM ATP, 300 mM ammonium acetate and 1 mM β-mercaptoethanol) and 350 mM Imidazole. Then, the protein was incubated with TEV and dialyzed against G-buffer (10 mM HEPES pH 7.5, 0.2 mM CaCl₂, 0.5 mM ATP, and 0.5 mM TCEP) and 300 mM ammonium acetate, overnight at 4˚C. After cleavage, the protein was loaded through a Ni-NTA column before a size-exclusion chromatography using Superdex 200 increase 10/300 GL column (GE Healthcare) in G-buffer. Peak fractions were pooled and concentrated. The ammonium acetate was removed using a PD SpinTrap G-25 column (GE Healthcare); the sample was eluted in G-buffer without ammonium acetate.

## Cryo-EM sample preparation, data collection, and image processing

Actin II filaments were polymerized with 1xF buffer and 0.01% sodium azide, both with and without JAS, which was used in a molar ratio of 1:2 (actin:JAS). Actin II filaments in the presence of JAS were diluted to 10 μM and those without JAS to 5 μM. CryoEM grids of these samples were prepared by the application of 3 μl to airglow discharged 200mesh copper Quantifoil R2/2 grids, which were then plunge frozen into liquid ethane using an MkIII Vitrobot (Thermofisher) operated at 90% humidity at 4˚C.

For actin II with JAS, 4133 movies were collected using a Titan Krios electron microscope equipped with a Falcon 3 camera, operated at 300 kV. The magnification was 75000x, corresponding to 1.09 Å pixel⁻¹. The movie stacks (46 frames) were aligned with MotionCor2 [63], and contrast transfer function parameters were estimated using CTFFind 4.1.10 [64]. The resulting 3977 aligned good micrographs were used for data processing. Filaments were picked with Relion AUTOPICK using a minimum inter-box distance of 30.5 Å, resulting in ~448000 particles, which were further classified with 3 rounds of reference-free 2D classification with 3x and, in the last round, with 2x binned particles. The resulting ~272000 good particles were reconstructed in Relion 3.1b. Refined half-maps were sharpened automatically [65], and the global resolution was corrected for the effects of a mask [66] using the Relion postprocessing

tool. The local resolution was estimated using Blocres in the Bsoft package version 1.8.6 with a Fourier shell correlation threshold of 0.143 [67,68].

For actin II without JAS, 1156 movies (46 frames) were collected as described above for the sample with JAS, and after estimating contrast transfer parameters using CTFFind 4.1.10, 1058 micrographs were retained. Using Relion 3.0.7 AUTOPICK with a minimum inter-box distance of 61 Å, 49102 filament particles were picked. After performing 2D reference-free classification, particles were re-extracted and classified with Relion 3D classification. The resulting 47197 particles were reconstructed, and contrast transfer function (CTF) parameters were refined. A final polishing step and final map reconstruction were performed in Relion 3.1b. Refined half-maps were sharpened automatically [62], and the global resolution was corrected for the effects of a mask [63] using the Relion postprocessing tool. The local resolution was estimated using Blocres in the Bsoft package version 1.8.6 with a Fourier shell correlation threshold of 0.143 [64,65].

## Model building and refinement

We built an atomic model of actin II based on a cryo-EM structure of actin I (5OGW) [24]. The actin I model was mutated using the actin II sequence and placed in the density map using Coot 0.9-pre [69] and Chimera 1.14 [70]. Namdinator was used to fit the initial atomic model to the density map and correct the model geometry [71]. The refinement was carried out for several rounds in reciprocal space with Phenix 1.19.2 [72].

## Structure analysis

The figures and videos of the structures were generated in Chimera 1.14 [70] and ChimeraX 1.2.5. The mass centers of the subdomains were calculated in Pymol v1.7.4, and the distances d2-4, b2, and the θ angle were computed with Chimera 1.14. The electrostatic potentials and hydrophobicity surface were generated in Chimera 1.14. The density map was segmented using ChimeraX 1.2.5 [73].

## Generation of actIIh73q

Point mutation H73Q was generated using the QuikChange II Site-directed mutagenesis kit from Agilent according to the manufacturer's instructions. The template for the generation by PCR of the desired mutation was the pSD141 plasmid containing the WT *actin II* locus. The plasmid was sequenced to verify the correct mutation. Transfection, cloning and genotyping was carried out as described above.

## Mass spectrometry

Zygotes from 1.5 h culture were isolated by magnetic beads coated with 13.1 antibody against Pbs21. Three samples were pooled in each experiment. The samples were lysed in 50 mM Tris-HCl, pH 7.8, 1 mM $CaCl_2$, 0.5 mM ATP [74] to which cocktails of protease and phosphatase inhibitors were added (PMSF, protease inhibitor cocktail from Sigma (cat no P2714) and phosphatase inhibitors A and B (from SIGMA)) with sonication 4 times 15 sec on ice. TritonX-100 was added to a final concentration of 1% and the samples were incubated for 30 min, followed by centrifugation at 8000 g for 15 min. The supernatant (30 μl) was loaded on a NuPAGE Bis-Tris 4–12% polyacrylamide gel (Invitrogen, Massachusetts, United States), and run in MOPS buffer at 200 V. The gel was then treated with Imperial Coomassie Stain (Thermo Fisher Scientific, CA, United States) and a coloured band below 42 kDa was cut, destained with a solution containing 50 mM bicarbonate ammonium (AMBIC) and acetonitrile (1:1), treated with 10

mM DTT at 56˚C for 45min, washed, incubated with 55 mM iodoacetamide at RT for 30 min, dried with acetonitrile, then in a Speed Vac, and treated with 12.5 ng/µl AspN (Promega Corporation, WI, United States) in 25 mM AMBIC overnight at 37˚C.

The peptide mixture obtained was analyzed by liquid chromatography-mass spectrometry (LC-MS/MS) analysis using an Ultimate 3000 HPLC (Dionex, Thermo Fisher Scientific) on-line with an Orbitrap Fusion Tribrid (Thermo Fisher Scientific, CA, United States) mass spectrometer. Peptides were desalted on a trap column (Acclaim PepMap 100 C18, Thermo Fisher Scientific) and then separated onto a 16 cm long silica capillary (Silica Tips FS 360-75-8, New Objective, MA, United States), packed in-house with a C18, 1.9 µm size particle (Michrom BioResources, CA, United States). A 60 min gradient was applied using buffer A (95% water, 5% acetonitrile, and 0.1% formic acid) and B (95% acetonitrile, 5% water, and 0.1% formic acid): B buffer was increased from 5% to 32% in 35 min, then to 80% in 5 minutes, and the column was washed and equilibrated again for the following run. MS spectra were acquired in the orbitrap at 120k while MS/MS spectra were acquired in the linear ion trap.

Spectra were analyzed using the software Proteome Dicoverer 2.4 (Thermo) using database containing *P. berghei* actin I and II sequences. Parameters used for the searches were: 15 ppm and 0.6 Da tolerance for precursor and fragment ions, respectively, fixed cysteine carbamido-methylation, variable methionine oxidation, histidine and lysine acetylation or methylation. Fixed Value PSM Validator node was applied taking into account maximum 0.05 deltaCn and high confidence PSMs (peptides spectral matches) with Xcorr = 2.5 and 3 for peptide charge 2 and 3, respectively.

## Sedimentation assay

Actin II was polymerized for 16 h at 4˚C in F buffer with and without JAS at a final concentration of 4 µM. The samples were centrifugated at 100000 $g$ to 435000 $g$ for 1 h at room temperature. Pellet or precipitated supernatant fractions were separated and analyzed by native-PAGE and SDS-PAGE. The intensity of the bands of pellet and supernatant fractions was measured using ImageLab (Biorad). The intensity values were normalized and plotted in GraphPad Prism 9.1.1.

## Dynamic light scattering

DLS was performed to analyze the *Plasmodium* actins oligomeric state of the proteins before and after polymerization and in the presence and absence of JAS. The samples at 1.3 mg/ml were measured using the Zetasizer Nano ZS (Malvern Instruments) in a final volume of 70 µl at 4˚C.

## Negatively stained electron micrographs

Actin II was polymerized for 16 h at 4˚C in F-buffer at a final concentration of 20 µM. The sample was then diluted to 1 µM in F buffer, 3 µl were applied on carbon-coated 200-mesh Cu grids (Electron Microscopy Sciences, Hatfield, PA). After 60 s of incubation, the grids were dried with Whatman paper and washed 3 times with F-buffer. The samples were stained with 2% uranyl acetate. The grids were imaged using a JEOL JEM-1230 microscope (JEOL Ltd., Tokyo, Japan) operated at 80 kV and with a final pixel size of 1.22 nm [29].

## Critical concentration determination

Different concentrations of *Plasmodium* actins were polymerized, as was explained before. The fluorescence was then measured from triplicate samples after 1 and 20 h at 4˚C. The plate

reader and the optical settings were the same as in the polymerization assays. The results were fitted using a segmental linear equation and 1/Y2 weighting in GraphPad Prism 9:

1. Y1 = intercept1 + slope1*X

2. YatX1 = slope1*X1 + intercept1

3. Y2 = YatX1 + slope2*(X–X1)

4. Y = IF(X<X1, Y1, Y2)

The first line of the equation corresponds to the first line segment. The second line of the equation calculates the Y value of the first regression when X = X1. The third line estimates the second regression segment. The final line defines Y for all values of X.

### Polymerization assays

The actin polymerization assay was performed as described before [33]. The reaction mixture contained 50 μl label and unlabeled actin with pyrene (in a ratio of 1:3) in GF-buffer with final concentrations between 0 to 7 μM, respectively. The reaction was started by adding 100 μl of 1x F-buffer (4 mM $MgCl_2$, 50 mM KCl and 10 mM EGTA pH 8). The final volume of the reaction was 150 μl. For nucleated polymerization assays, actin II seeds were added to the reaction mixture at a final concentration of 0.5 μM. The fluorescence of nucleated and non-nucleated polymerization assays was monitored for 2 h for actin I and actin II at 20˚C. Measurements were carried out in a Tecan Spark 20M multimode microplate reader using black 96-well plates (Greiner), λex = 365 nm (9 nm bandpass) and λem = 407 nm (20 nm bandpass), 5 flashes per measurement and a 5-s orbital mixing step performed at 216 rpm before commencing the measurements. The baseline of polymerization curves was corrected and normalized against the maximum value of actin I or actin II in GraphPad Prism 9.1.1.

### Supporting information

**S1 Table. Data collection and refinement statistics.**
(DOCX)

**S2 Table. Relative kinetic parameters of actin II polymerization.** All values are reported as mean ± standard deviation (actin II seeds: n = 3 NP, n = 5 in SS). * Relative k. calculated using the steady state $C_c$ and relative elongation constant ($k_+$) from slope 2 of nucleated polymerization assays. Raw data related to the table can be found in S7 Data.
(DOCX)

**S3 Table. Primers used for DNA constructs and genotyping.**
(DOCX)

**S1 Fig. (a)** Genotyping *flag-actII* cloned line. Left integration of the construct was verified with primer pair A2F2 and A2R and right integration with the primer pair DHFR and mCherryR. To control for absence of the *actII(-)* parasites the primer pair A2F1 and mCherryR was used. Lane 1 and 2: *flag::actII*, lane 1, 135 ng, lane 2, 50 ng template; lane 3: *actII(-)*. Quality control of gDNA used the *gapdh* primer pair of the same samples. **(b)**. **(b-e)** Phenoptyic analysis of *flag::actII* compared to WT. **(b)** Exflagellation analysis; average of three experiments of the WT and four of the mutant. **(c)** Ookinete conversion; three experiments of each strain. Error bars in **(b)** and **(c)** are S.E.M. **(d)** Number of oocysts/midgut. The oocysts were stained with Cap380 antibody and all oocysts irrespective of size are plotted. Differences in **(b-d)** are not significant, Student's t-test for **(b)** and **(c)**, Mann-Whitney for **(d)** and **(e)**. Raw data related

to panels b-d can be found in S8 Data.
(TIF)

**S2 Fig. (a)** Western blot of extracts from midguts infected with the line expressing FLAG::acti-nII. Midguts were dissected on day 3 and 10 post blood feeding and crude extracts of 20 and 13 midguts, respectively, were loaded in each lane. Right panel is a positive control with zygote extracts. The samples were run on the same gel and the blot was probed with the anti-FLAG antibody. A duplicate blot was probed with anti-CSP antibody as a loading control; it only gave a signal for the day 10 sample. **(b,c)** Immunolabeling of *flag::actII* female gamete 8 min p. a. **(b)** and ookinete **(c)**. No signal was detected with the anti-FLAG antibody. The background GFP signal (green) is constitutively expressed in this line. DNA was stained with Hoechst 33342 (blue). Scale bars, 5 μm. The female gamete originates from the same experiment as the 8 min sample of male gametes in **Fig 1** (lower panels).
(TIF)

**S3 Fig.** Evolutionary conservation of *P. falciparum* actin II in comparison with **(a)** actin I from *Plasmodium* spp. and **(b)** skeletal muscle α-actins. The actin surface is colored according to conservation scores; high (purple) to low (green). Amino acid conservation was estimated using the ConSurf server [75].
(TIF)

**S4 Fig.** Fourier shell correlation plot of actin II and JAS-stabilized actin II. The corrected curve was calculated from independently refined half-datasets with a soft mask filtered to 15 Å in Relion [76].
(TIF)

**S5 Fig.** Local resolution of actin II **(a)** and JAS-stabilized actin II **(b)**. The local resolution esti-mation is based on Fourier shell correlation threshold 0.143 calculated with Blocres in the Bsoft software package, applied to the final sharpened map. The left panel shows a central sec-tion of the filament. On the right, two adjacent protomers are shown and, the ligands densities are highlighted [68].
(TIF)

**S6 Fig.** The distance of the twist angles of the centers of mass of SDs ($\theta$), the phosphate clamp distance (b2), the distance of SD1 and SD2 (d2-4), SD3-SD4 (d3-4), and cleft mouth (c) calcu-lated for different actin structures. The dihedral angle of subdomains in the JAS-stabilized actin II model is rotated 11.8˚ relative to the crystal structure of G-actin II-gelsolin (PDB: 6I4M), 2.4˚ relative to JAS stabilized F-actin I (5OGW), and 3.4˚ relative to the JAS stabilized F-actin I in complex with myosin A (6TU4).
(TIF)

**S7 Fig. C termini of F-actin II and JAS-stabilized F-actin II. H372 interacts with E117. (a)** In F-actin II, the ring of H372 is oriented towards the E117. **(b)** In F-actin II, the N1 of the H372 turns towards K113. The distance between H372 and K113 is more than 5 Å in both structures.
(TIF)

**S8 Fig. Intrastrand and interstrand interactions of F-actin II and other filamentous actin structures [11,24–26].** Two protomers are represented by spheres n-1 and n-2 (salmon and pink); the silhouette of the third protomer (n) is visualized by ribbon representation in Chi-mera.
(TIF)

**S9 Fig. Intrastrand contacts near the JAS-binding site.** (a) *P. falciparum* F-actin II, (b) JAS-stabilized F-actin II, (c) *P. falciparum* F-actin I (6TU4), (d) filamentous skeletal muscle α-actin (6DJO). In actin II, I288 inserts into a groove in the adjacent protomer, resembling a lock-key interaction, like in canonical actins. The actin surface is colored according to hydrophobicity; high (yellow), medium (white), and low (purple).
(TIF)

**S10 Fig.** Lateral contacts of protomers from a different strand of (a) F-actin II-JAS, (b) F-actin I (6TU4), (c) *O. cuniculus* F-actin (6DJO). Distances between the residues are indicated in black dashed lines.
(TIF)

**S11 Fig. MS/MS spectra indicating methylation of H73 in *Plasmodium* actin II.** (a) *Pb*ActII peptide 56–79. The red stars indicate methylated residues. (b) Recombinant PfActII peptide 69–84. Methylated H73 is indicated by a green star. Methylated double and triple-charged signals are highlighted by green boxes. The b and y series ions detected are shown in red and blue, respectively, in both panels. The m/z difference between these ions corresponds to a methylated histidine. Raw data related to this figure can be found in <u>S9 Data</u>.
(TIF)

**S12 Fig. Relative abundance of methylated and unmethylated peptides (residues 69–84) in recombinant *Pf*ActII and skeletal muscle α-actin.** (a) Unmethylated and (b) methylated *Pf*ActII. (c) Unmethylated and (d) methylated skeletal muscle α-actin. Raw data related to this figure can be found in <u>S9 Data</u>.
(TIF)

**S13 Fig. The orientation of the A-loop in actin II.** (a) F-actin II, (b) F-actin II-JAS, (c) F-actin I-JAS (6TU4) and (d) *G. gallus* F-actin (6DJO). All structures are in the Mg-ADP state and show a 1b conformation of the A-loop. The most probable ionic and hydrogen bonds are indicated with dashed lines.
(TIF)

**S14 Fig. Genotyping of *actIIh73q*. Left integration of the construct was verified with primer pair A2F2 and A2R and right integration with the primer pair DHFR and mCherryR.** To control for absence of the actII(-) parasites the primer pair A2F1 and mCherryR was used. **1:** WT; **2:** positive control; **3:** transfected parasites with actIIh73q construct; **4 and 5:** actIIh73q clone 1. For quality control of the gDNA GAPDH primer pair was used.
(TIF)

**S15 Fig. Two steps of sedimentation assay.** (a) Native PAGE of *Plasmodium* actins. Samples were polymerized overnight, pellet (P) and supernatant (SN) were separated by ultracentrifugation at 4˚C for 1 h (UC1). The SN fraction was re-pelleted 16 h after the initial ultracentrifugation (UC2). (b) SDS-PAGE of the two steps of ultracentrifugation at 435000 *g* of actin II.
(TIF)

**S16 Fig. D-loop conformation in actin II. Superimposed structures of JAS-stabilized F-actin II, G-actin II, JAS-stabilized *O. cuniculus* F-actin (5OOC), and *O. cuniculus* F-actin (5ONV).** JAS is not altering the conformation of the D-loop of actin II like in actin from *O. cuniculus*, in which the D-loop has an open conformation.
(TIF)

**S17 Fig. $C_c$ plots of actin II ($n^b = 2$). a.** Samples after 1 h of incubation at 4˚C. **b.** Samples after 20 h of incubation at 4˚C. The data in both panels fit in a linear regression equation. X0

represents the critical concentration. The lower concentration points are magnified on the left. Error bars represent the standard deviation, a.u. = arbitrary units. $n^b$ = biological replicates with 3 technical replicates per experiment. Raw data related to this figure can be found in S5 Data.
(TIF)

**S1 Movie. Evolutionary conservation of *P. falciparum* actin II in comparison with A actin I from *Plasmodium* spp.** The actin surface is colored according to conservation scores; high (purple) to low (variable). Amino acid conservation was estimated using the ConSurf server [75].
(MOV)

**S2 Movie. Twistedness motion of G-actin II and F-actin II protomers.** G-actin II (6I4M) structure and our F-actin II model were used to generate the motion using Morph conformation in Chimera [70]. There is no crystal structure of actin II available, G-actin from *P. berghei* shares 92.6% sequence identity with actin II. The movement of residues around the nucleotide-binding site is shown.
(MOV)

**S3 Movie.** The twist angles of the mass centers of SDs ($\theta$) of F-actin II protomers. Superimposed structures of JAS-stabilized F-actin II (medium purple) with **(a)** G-actin II (PDB: 6I4M) (lemon) and **(b)** actin I in F form (6TU4) (light blue). The spheres represent the center mass of each subdomain. The center of mass was calculated in Pymol v1.7.4 and visualized in Chimera [70]. Structures are aligned using SD3. F, filament, and G, globular.
(MOV)

**S1 Data. Raw data file for Fig 5A and 5B.**
(XLSX)

**S2 Data. Raw data file for Fig 6A, 6B, 6C and 6D.**
(XLSX)

**S3 Data. Raw data file for Fig 7B.**
(PZFX)

**S4 Data. Raw data file for Fig 7C and 7D.**
(XLS)

**S5 Data. Raw data file for Figs 8A, 8B, 8C, 8D and S17.**
(XLSX)

**S6 Data. Raw data file for Fig 8E and 8F.**
(XLSX)

**S7 Data. Raw data file for Fig 8G and 8H and S3 Table.**
(XLSX)

**S8 Data. Raw data file for S1 Fig.**
(XLSX)

**S9 Data. Raw data file for S11 and S12 Figs.**
(XLSX)

## Acknowledgments

Marialuisa Casella provided valuable technical assistant in the mass spectrometry analysis. We thank the Francis Crick Structural Biology Scientific Technology Platform for instrument access and computing support: Andrea Nans for help with data cryo-EM data collection and Phil Walker, Andy Purkiss, and Andrea Nans for help with computing.

## Author Contributions

**Conceptualization:** Juha Vahokoski, Inga Sidén-Kiamos, Inari Kursula.

**Formal analysis:** Andrea J. Lopez, Maria Andreadaki, Peter B. Rosenthal, Inga Sidén-Kiamos, Inari Kursula.

**Funding acquisition:** Peter B. Rosenthal, Inga Sidén-Kiamos, Inari Kursula.

**Investigation:** Andrea J. Lopez, Maria Andreadaki, Juha Vahokoski, Elena Deligianni, Lesley J. Calder, Serena Camerini, Anika Freitag, Ulrich Bergmann, Peter B. Rosenthal, Inga Sidén-Kiamos, Inari Kursula.

**Methodology:** Andrea J. Lopez, Maria Andreadaki, Elena Deligianni, Lesley J. Calder, Serena Camerini, Anika Freitag, Ulrich Bergmann.

**Project administration:** Peter B. Rosenthal, Inga Sidén-Kiamos, Inari Kursula.

**Resources:** Peter B. Rosenthal.

**Supervision:** Juha Vahokoski, Peter B. Rosenthal, Inga Sidén-Kiamos, Inari Kursula.

**Validation:** Peter B. Rosenthal, Inga Sidén-Kiamos, Inari Kursula.

**Visualization:** Andrea J. Lopez, Maria Andreadaki, Inga Sidén-Kiamos.

**Writing – original draft:** Andrea J. Lopez, Inga Sidén-Kiamos.

**Writing – review & editing:** Andrea J. Lopez, Juha Vahokoski, Peter B. Rosenthal, Inari Kursula.

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
