## [Decision Letter · Decision Letter 0]

21 Jan 2023

Dear Dr. Kursula,

Thank you very much for submitting your manuscript "Structure and function of Plasmodium actin II in the parasite mosquito stages" for consideration at PLOS Pathogens. As with all papers reviewed by the journal, your manuscript was reviewed by members of the editorial board and by several independent reviewers. The reviewers appreciated the attention to an important topic. Based on the reviews, we are likely to accept this manuscript for publication, providing that you modify the manuscript according to the review recommendations.

Sincerely,

Matthew K. Higgins

Academic Editor

PLOS Pathogens

Dominique Soldati-Favre

Section Editor

PLOS Pathogens

Kasturi Haldar

Editor-in-Chief

PLOS Pathogens

orcid.org/0000-0001-5065-158X

Michael Malim

Editor-in-Chief

PLOS Pathogens

orcid.org/0000-0002-7699-2064

Reviewer Comments (if any, and for reference):

Reviewer's Responses to Questions

**Part I - Summary**

Reviewer #1: In this excellent paper the authors combine high resolution structural analysis with reverse genetics to investigate actin II in Plasmodium parasites. Plasmodium has a ubiquitously expressed actin I and a divergent actin II expressed in the mosquito stages. Both actins are essential for the parasite and of high interest to understand actin biology across eukaryotic life. While structures and several transgenic lines expressing mutant versions are available for actin I, the current paper investigates both for actin II.

Reviewer #2: The authors reported spatio-temporal distribution in cells, as well as biochemistry and structure, of Plasmodium actin II. The detailed features of Plasmodium actin II achieved by well-organized experimental procedures should be of benefit to a wide range of researchers in the field of pathogens and cytoskeletons. I believe this manuscript deserves publication in PLOS Pathogens.

Reviewer #3: The manuscript by Lopez et al presents a detailed structural and biophysical characterisation of actin II from Plasmodium falciparum.

They start by showing a different expression pattern within the Plasmodium life cycle for actin II compared with actin I, with most expression in gametocytes, with actin mostly found in the nucleus, unlike actin I. This indicates a likelihood of divergent functions for actin II vs actin I. This seems convincing and is largely well done, albeit with challenges to understand figure 1 due to lack of labelling.

They next determine the structure of actin II filaments using cryo-EM methods. This is well done and is accompanied by an extensive discussion and comparison with other actins, including Plasmodium actin I.

They next conduct an experiment in which they introduce a stabilised form of actin I into the actin II locus and show that it does not complement. I was not sure about the significance of this experiment, but the authors make an appropriately conservative interpretation that other features of actin II are also important, which seems sensible.

The authors then identify methylation of actin II and so that the methylated residue is important for function, and they explore the polymerisation behaviour and dynamics of actin II.

In summary, this is a useful manuscript, which presents the first structural and detailed biophysical characterisation of Plasmodium actin II and identifies features, in expression pattern, localisation and biophysics through which it differs from actin I. The manuscript doesn’t identify a novel function of actin II, but it is still a valuable contribution to the field and worthy of publication.

**Part II – Major Issues: Key Experiments Required for Acceptance**

Reviewer #1: no experiments required, there is great scope for future work, but please elaborate/speculate in discussion:

could there be a gene expression difference in the H73 mutant? If not, how else can the “late” phenotype in oocysts be explained?

Reviewer #2: (No Response)

Reviewer #3: (No Response)

**Part III – Minor Issues: Editorial and Data Presentation Modifications**

Reviewer #1: I hope the following helps to improve an already phenomenal paper:

37: “We show… zygotes…” – maybe better to write “We confirm expression and how function”, also: I would suggest to add function during “oocyst stage”

44/105/…: I understand that paper such that actin II is expressed in zygote but not oocysts, but that there is a functional deficiency in oocysts and not zygotes. If correct, please modify to state “oocyst” instead of “zygote” when talking about function.

79/80: modify to say “development, motility and invasion” and cite the work by Douglas (ie ref 49 and Yee et al., Plos Path 2022) to qualify “intensely studied” and the work by Das et al BMC Bio 2017 to qualify “development and invasion”.

182: I don’t see how localization offers new insight into function, please rewrite or delete

193: add a sentence on how you purified actin II

208 and onwards: please use sensible digits, e.g 167 instead of 166,9°

313: delete first “and”

646: not sure “filament like structures” is the best way to describe the accumulations

Reviewer #2: 1 It is difficult to follow overall structural difference among the subunits of PfActII, PfActI and alpha-actin. A superposed diagram of one subunit of each species such as Fig. S16 would be helpful.

2 Fig. 1 presented spatio-temporal distribution of actin II in the cell. I agree with the authors that the rod-like distribution imply filament formation. However, staining actin II with rhodamine phalloidin or lifeact, which binds to the actin filaments, will give more clarity.

3 Line 269: “(Fig 4a-5c)” should be “(Fig 4a-4c)”.

4 Lines 375-376: “Under polymerizing conditions, actin II was distributed into three populations with rH of 8.1, 43, and 410 nm, with volume percentage contributions of 51, 33, and 17%.” I believe the authors could observe images corresponding to the fraction with rH of 43 nm by electron microscopy. The authors should report it. The state of this fraction might be largely different from the filament and it might cause the two linear slopes in Figs 8c and d.

5 Lines 550-551: “In actin II, fragmentation could be mediated in vivo by other mechanisms, such as ABPs or the intracellular environment.” The authors should cite papers showing evidences for fragmentation of actin II filaments in vivo.

6 Lines 1321-1322: I could not find “green star” in Fig S11a.

Reviewer #3: • A number of the figures are not particularly well labelled or explained and would benefit from work to make the manuscript more accessible to a reader.

• Please adapt the labelling of Figure 1a to make it easier to read – for example label the lanes with their contents rather than just numbers which require the legend to interpret.

• Please rewrite the legend to Figure 1b-e, which is very confusing and unclear. Which images have just the secondary control? What are c and e, which have no labelling at all? The figure needs labelling properly.

• The entire section from lines 207-231 is only describing supplementary figures. If important enough for a full section of the text, then some of the data could be shown in panels in Figure 2 in the main manuscript.

• Figure 3 – these isn’t a good match to the text section of lines 244-257. For example, GgAct isn’t described in the text. The same is true when comparing lines 262-273 with Figure 4. Please join up the figures and the text.

• Figure 5C – what is the difference between the central and right-hand columns?

• Figure 7B – why has the PfAcI not been studied at 100k?

• Figure 7C – label with arrows the species described in the text?

• Figure 8 – why is actin I studied after 1h and atcin II after 20h?

• Finally, have the coordinates and cryo-EM data been deposited. There were no validation reports provided. The manuscript shouldn’t be accepted without these being provided for the reviewers to check.

PLOS authors have the option to publish the peer review history of their article (what does this mean?). If published, this will include your full peer review and any attached files.

Reviewer #1: No

Reviewer #2: **Yes: **Akihiro Narita

Reviewer #3: No

Figure Files:

Data Requirements:

Reproducibility:

References:

---

## [Editor Report · Decision Letter 1]

3 Feb 2023

Dear Dr. Kursula,

We are pleased to inform you that your manuscript 'Structure and function of Plasmodium actin II in the parasite mosquito stages' has been provisionally accepted for publication in PLOS Pathogens.

Best regards,

Matthew K. Higgins

Academic Editor

PLOS Pathogens

Dominique Soldati-Favre

Section Editor

PLOS Pathogens

Kasturi Haldar

Editor-in-Chief

PLOS Pathogens

orcid.org/0000-0001-5065-158X

Michael Malim

Editor-in-Chief

PLOS Pathogens

orcid.org/0000-0002-7699-2064
---

## [Editor Report · Acceptance letter]

1 Mar 2023

Dear Dr. Kursula,

We are delighted to inform you that your manuscript, "Structure and function of Plasmodium actin II in the parasite mosquito stages," has been formally accepted for publication in PLOS Pathogens.

Best regards,

Kasturi Haldar

Editor-in-Chief

PLOS Pathogens

orcid.org/0000-0001-5065-158X

Michael Malim

Editor-in-Chief

PLOS Pathogens

orcid.org/0000-0002-7699-2064